# HOLISTICALLY EXPLAINABLE VISION TRANSFORMERS

## ABSTRACT

Transformers increasingly dominate the machine learning landscape across many tasks and domains, which increases the importance for understanding their outputs. While their attention modules provide partial insight into their inner workings, the attention scores have been shown to be insufficient for explaining the models as a whole. To address this, we propose B-cos transformers, which inherently provide *holistic* explanations for their decisions. Specifically, we formulate each model component—such as the multi-layer perceptrons, attention layers, and the tokenisation module—to be *dynamic linear*, which allows us to faithfully summarise the entire transformer via a single linear transform. We apply our proposed design to Vision Transformers (ViTs) and show that the resulting models, dubbed **Bcos-ViTs**, are highly interpretable and perform competitively to baseline ViTs on ImageNet. Code will be available: github.com/anonymous/authors.

## 1 INTRODUCTION

Convolutional neural networks (CNNs) have dominated the last decade of computer vision. However, recently they are often surpassed by transformers (Vaswani et al., 2017), which—if the current development is any indication—will replace CNNs for ever more tasks and domains. Transformers are thus bound to impact many aspects of our lives: from healthcare, over judicial decisions, to autonomous driving. Given the sensitive nature of such areas, it is of utmost importance to ensure that we can explain the underlying models, which still remains a challenge for transformers.

To explain transformers, prior work often focused on the models' attention layers (Jain & Wallace, 2019; Serrano & Smith, 2019; Abnar & Zuidema, 2020; Barkan et al., 2021), as they inherently compute their output in an interpretable manner. However, as transformers consist of many additional components, explanations derived from attention alone have been found insufficient to explain the full models (Bastings & Filippova, 2020; Chefer et al., 2021). To address this, our goal is to develop transformers that inherently provide *holistic* explanations for their decisions, i.e. explanations that reflect *all* model com-

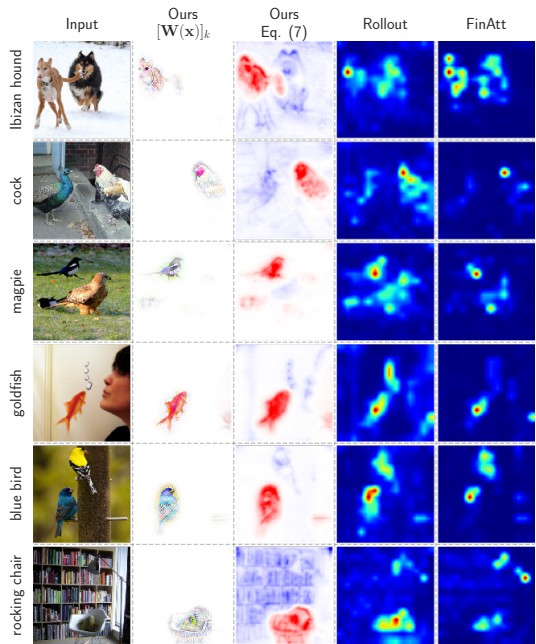

Fig. 1: Inherent explanations (cols. 2+3) of B-cos ViTs vs. attention explanations (cols. 4+5) for the same model. Note that $\mathbf{W}(\mathbf{x})$ faithfully reflects the *whole* model and yields more detailed and class-specific explanations than attention alone. For a detailed discussion, see supplement.

ponents. These model components are given by: a tokenisation module, a mechanism for providing positional information to the model, multi-layer perceptrons (MLPs), as well as normalisation and attention layers, see Fig. 2a. By addressing the interpretability of each component individually, we obtain transformers that *inherently* explain their decisions, see, for example Fig. 1 and Fig. 2b.

In detail, our approach is based on the idea of designing each component to be *dynamic linear*, such that it computes an input-dependent linear transform. This renders the entire model dynamic linear, cf. Böhle et al. (2021; 2022), s.t. it can be summarised by a single linear transform for each input.

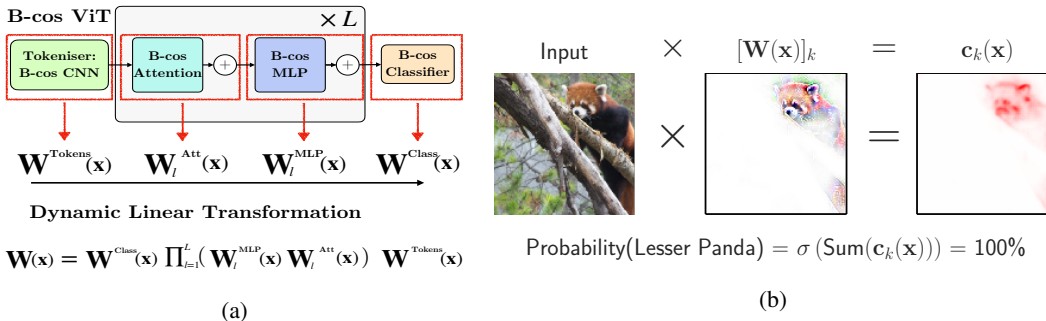

Fig. 2: **(a) B-cos ViTs.** We design each ViT component to be dynamic linear, allowing us to summarise the entire model by a single linear transform $\mathbf{W}(\mathbf{x})$, as shown in the bottom. **(b) Computation is Explanation.** The model output is exactly computed by the linear transform $\mathbf{W}(\mathbf{x})$. As a result, we can visualise this effective linear transform either by the corresponding matrix row (center) or the contributions $\mathbf{c}_k(\mathbf{x})$ (right), cf. Eq. (7).

In short, we make the following contributions. **(I)** We present a novel approach for designing inherently interpretable transformers. For this, **(II)** we carefully design each model component to be dynamic linear and ensure that their combination remains dynamic linear and interpretable. Specifically, we address **(IIa)** the tokenisation module, **(IIb)** the attention layers, **(IIc)** the MLPs, and **(IId)** the classification head. **(III)** Additionally, we introduce a novel mechanism for allowing the model to learn attention priors, which breaks the permutation invariance of transformers and thus allows the model to easily leverage positional information. In our experiments, we find that B-cos ViTs with such a learnt 'attention prior' achieve significantly higher classification accuracies. **(IV)** Finally, we evaluate a wide range of model configurations and show that the proposed B-cos ViTs are not only highly interpretable, but also constitute powerful image classifiers.

## 2  RELATED WORK

**Attention as Explanation.** As the name exemplifies, attention is often thought to give insight into what a model 'pays attention to' for its prediction. As such, various methods for using attention to understand the model output have been proposed, such as visualising the attention of single attention heads, cf. Vaswani et al. (2017). However, especially in deeper layers the information becomes increasingly distributed and it is thus unclear whether a given token still represents its original position in the input (Serrano & Smith, 2019; Abnar & Zuidema, 2020), thus complicating the interpretation of high attention values deep in the network (Serrano & Smith, 2019; Bastings & Filippova, 2020).

Therefore, Abnar & Zuidema (2020) proposed 'attention rollout', which summarises the various attention maps throughout the layers. However, this summary still only includes the attention layers and neglects all other network components (Bastings & Filippova, 2020). In response, various improvements over attention rollout have been proposed, such as GradSAM (Barkan et al., 2021) or an LRP-based explanation method (Chefer et al., 2021), that were designed to more accurately reflect the computations of *all* model components. The significant gains in quantitative interpretability metrics reported by Chefer et al. (2021) highlight the importance of such holistic explanations.

Similarly, we also aim to derive *holistic* explanations for transformers. However, instead of deriving an explanation 'post-hoc' as in Chefer et al. (2021), we explicitly design our models to be holistically explainable. For this, we formulate each component—and thus the full model—to be dynamic linear.

**Dynamic Linearity.** Plain linear models, i.e. $y(\mathbf{x}) = \mathbf{W}\mathbf{x}$, are usually considered interpretable, as $y(\mathbf{x})$ can be decomposed into individual contributions $c_i = w_i\,x_i$ from any dimension $i$: $y = \sum_i c_i$ (Alvarez-Melis & Jaakkola, 2018). However, linear models have a limited capacity, which has lead to various works aimed at extending their capacity without losing their interpretability, see Alvarez-Melis & Jaakkola (2018); Brendel & Bethge (2019); Böhle et al. (2021; 2022). An appealing strategy for this is formulating *dynamic* linear models (Alvarez-Melis & Jaakkola, 2018; Böhle et al., 2021; 2022), i.e. models that transform the input with a data-dependent matrix $\mathbf{W}(\mathbf{x})$: $y(\mathbf{x}) = \mathbf{W}(\mathbf{x})\mathbf{x}$.

In this work, we rely on the B-cos framework (Böhle et al., 2022), but instead of focusing on CNNs as in Böhle et al. (2022), we investigate the applicability of this framework to transformers.

**Interpretability in DNNs.** The question of interpretability extends, of course, beyond transformers and many methods for explaining DNNs have been proposed. While other approaches exist, cf. Kim

et al. (2018), these methods typically estimate the importance of individual input features, which can be visualised as a heatmap, cf. Lundberg & Lee (2017); Petsiuk et al. (2018); Ribeiro et al. (2016); Simonyan et al. (2014); Springenberg et al. (2015); Zhou et al. (2016); Bach et al. (2015); Selvaraju et al. (2017); Shrikumar et al. (2017); Srinivas & Fleuret (2019); Sundararajan et al. (2017).

Similarly, our models yield explanations in form of contribution heatmaps. However, in contrast to the above-referenced post-hoc explanation methods, the contribution maps of our B-cos ViTs are *model-inherent*. Further, as the interpretability of the B-cos ViTs relies on aligning the weights with the inputs, the weights can be visualised in colour as in Böhle et al. (2022), see Figs. 1 and 7.

## 3 DESIGNING HOLISTICALLY EXPLAINABLE TRANSFORMERS

In the following, we present the overarching goal that we pursue and how we structure this section around it. First, however, we introduce the necessary background and the notation used in our work.

**Preliminaries.** Vision Transformers (ViT) (Dosovitskiy et al., 2021) with $L$ blocks are given by:

$$\mathbf{y}(\mathbf{x}) = \text{Classifier} \circ \prod_{l=1}^{L} (\text{MLPBlock}_l \circ \text{AttBlock}_l) \circ \text{Tokens}(\mathbf{x}). \quad (1)$$

Here, the input $\mathbf{x} \in \mathbb{R}^{(CHW)}$ denotes a vectorised image of $H = W$ height and width and with $C$ color channels; the functions, concatenated by $\circ$, are defined in Eqs. (2) - (5) (left), with $\mathbf{P}, \mathbf{E} \in \mathbb{R}^{D \times N}$, $N$ the number of tokens and $D$ their dimensionality. Further, in Eqs. (2) - (5), CNN is a convolutional neural network[1], $\mathcal{T}$ 'tokenises' the CNN output (see Sec. 3.2), Linear is a learnable linear transform $\mathbf{p}'(\mathbf{p}) = \mathbf{W}\mathbf{p} + \mathbf{b}$ with parameters $\mathbf{W}$ and $\mathbf{b}$ that is applied to each token $\mathbf{p}$ (columns of $\mathbf{P}$) independently, MSA denotes Multi-head Self-Attention, and $\mathbf{E}$ is a learnable embedding. Following Graham et al. (2021), Pool performs average pooling over the tokens, and the model output is given by $\mathbf{y}(\mathbf{x}) \in \mathbb{R}^M$ with $M$ classes. Last, we omit indices for blocks and layers whenever unambiguous and it may be assumed that each layer has its own set of learnable parameters.

**Our goal** in this work is to reformulate the ViTs such that they compute their output in a more interpretable manner. Specifically, we aim to make them *dynamic linear* such that they compute $\mathbf{y}(\mathbf{x}) = \mathbf{W}(\mathbf{x})\,\mathbf{x}$. Instead of using the ViTs to predict $\mathbf{W}(\mathbf{x})$, cf. Alvarez-Melis & Jaakkola (2018), we achieve this by rendering each of the ViT components dynamic linear on their own as follows:

$$\text{Tokens}(\mathbf{x}) = \mathcal{T}(\text{CNN}(\mathbf{x})) + \mathbf{E} \longrightarrow \text{B-cos Tokens} \quad (\mathbf{x}) = \mathbf{W}^{\text{Tokens}}(\mathbf{x})\,\mathbf{x} \quad (2)$$

$$\text{AttBlock}(\mathbf{P}) = \text{MSA}(\mathbf{P}) + \mathbf{P} \longrightarrow \text{B-cos AttBlock} \quad (\mathbf{P}) = \mathbf{W}^{\text{Att}} \quad (\mathbf{P})\,\mathbf{P} \quad (3)$$

$$\text{MLPBlock}(\mathbf{P}) = \text{MLP}(\mathbf{P}) + \mathbf{P} \longrightarrow \text{B-cos MLPBlock}\, (\mathbf{P}) = \mathbf{W}^{\text{MLP}} \quad (\mathbf{P})\,\mathbf{P} \quad (4)$$

$$\text{Classifier}(\mathbf{P}) = \text{Linear} \circ \text{Pool}(\mathbf{P}) \longrightarrow \text{B-cos Classifier} \quad (\mathbf{P}) = \mathbf{W}^{\text{Class}} \quad (\mathbf{P})\,\mathbf{P}. \quad (5)$$

Crucially, we define each component such that it can be expressed as a dynamic linear function, see the right-hand side of Eqs. (2) - (5). As a result, the entire model will become dynamic linear:

$$\mathbf{y}(\mathbf{x}) = \mathbf{W}^{\text{Class}}(\mathbf{x})\,\prod_{l=1}^{L}\left(\mathbf{W}_l^{\text{MLP}}(\mathbf{x})\,\mathbf{W}_l^{\text{Att}}(\mathbf{x})\right)\,\mathbf{W}^{\text{Tokens}}(\mathbf{x})\,\mathbf{x} = \mathbf{W}(\mathbf{x})\,\mathbf{x}. \quad (6)$$

Specifically, we develop the B-cos ViTs in accordance with the B-cos framework (Böhle et al., 2022) to render $\mathbf{W}(\mathbf{x})$ interpretable by aligning it with relevant input patterns.

**Outline.** In the following, we shortly summarise the most relevant aspects of B-cos networks and how to explain them (Sec. 3.1). Then, we discuss Eqs. (2) - (5) in detail and how we ensure that the resulting linear transform $\mathbf{W}(\mathbf{x})$ will be interpretable. In particular, we discuss the tokenisation (Sec. 3.2), attention (Sec. 3.3), and the multi-layer perceptrons (Sec. 3.4). Finally, in Sec. 3.5, we discuss how we encode positional information in B-cos ViTs, introducing 'position-aware' attention.

### 3.1 B-COS NETWORKS: INTERPRETABLE MODEL-INHERENT EXPLANATIONS

As shown in Eq. (6), a dynamic linear transformer is summarised *exactly* by a single matrix $\mathbf{W}(\mathbf{x})$ for every $\mathbf{x}$. These linear explanations lend themselves well for understanding the model decisions: as in plain linear models, one can calculate linear contributions from individual features (e.g., pixels) to each output unit. In detail, the effective linear contributions $\mathbf{c}_k$ to the $k$-th class logit are given by

**Dynamic Linear Contribution Maps:** $\quad \mathbf{c}_k(\mathbf{x}) = [\mathbf{W}(\mathbf{x})]_k^T \odot \mathbf{x}, \quad (7)$

---

[1]To simplify later equations, we take advantage of the fact that conv. layers are equivalent to linear layers with weight constraints (weight sharing and local connectivity) and assume the CNN to process vectorised images.

with $\odot$ denoting element-wise multiplication. Crucially, these contribution maps *faithfully* summarise the entire model, as this linear summary is inherent to the model formulation, see also Fig. 2. Thus, we use these contribution maps to explain the B-cos ViTs, see, e.g., Figs. 1, 2 and 7.

Note, however, that while the contribution maps in Eq. (7) accurately summarise any given dynamic linear model, this summary need not be *interpretable*. E.g., for piece-wise linear models, which are also dynamic linear, this amounts to 'Input×Grad', cf. Adebayo et al. (2018). For such models, however, the contributions $\mathbf{c}$ are very noisy and not easily interpretable for humans. Hence, we design the transformers in accordance with the 'B-cos' framework (Böhle et al., 2022), which ensures that $\mathbf{W}(\mathbf{x})$ aligns with relevant input features and thus becomes easily interpretable. In detail, the B-cos transform induces weight alignment by suppressing outputs for badly aligned weights:

$$\text{B-cos}(\mathbf{a}; \mathbf{W}) = \left( \cos^{\text{B}-1}(\mathbf{a}, \mathbf{W}) \odot \widehat{\mathbf{W}} \right) \mathbf{a} = \mathbf{W}(\mathbf{a})^T \mathbf{a} . \tag{8}$$

here, $\cos$ is applied row-wise, $\widehat{\mathbf{W}}$ denotes that the matrix rows are of unit norm, and $\odot$ represents row-wise scaling. As can be seen on the RHS of Eq. (8), the B-cos transform is dynamic linear.

As a result, the matrix rows $[\mathbf{W}(\mathbf{x})]_k$ align with relevant patterns of class $k$. By encoding the image such that the color is uniquely determined by the angle of the pixel encodings, it is possible to directly visualise those matrix rows in color, see Figs. 1 and 7. For details, see Böhle et al. (2022).

**Requirements.** To ensure that a B-cos network aligns $\mathbf{W}(\mathbf{x})$ with its inputs, each of its layers needs to **(a)** be bounded, **(b)** yield its maximum output if and only if its weight vectors align with its input, and **(c)** directly scale the overall *model* output by its own output norm, see Böhle et al. (2022). In the following, we ensure that each of the model components (Eqs. (2)-(5)), fulfills these requirements.

## 3.2 Interpretable Tokenisation Modules: B-cos CNNs

While the original Vision Transformer only applied a single-layered CNN, it has been shown that deeper CNN backbones yield better results and exhibit more stable training behaviour (Xiao et al., 2021). Hence, to address the general case, and to take advantage of the increased training stability and performance, we take the tokenisation module to be given by a general CNN backbone. Being able to explain the *full* ViT models consequently requires using an explainable CNN for this.

**Tokenisation.** We use B-cos CNNs (Böhle et al., 2022) as feature extractors; thus, the requirements **(a-c)**, see Sec. 3.1, of B-cos networks are, of course, fulfilled. The input tokens are computed as

$$\boxed{\text{B-cos Token } \mathbf{p}_i(\mathbf{x}) = \mathcal{T}_i \circ \text{B-cos CNN}(\mathbf{x}) = \mathbf{W}^{\mathcal{T}_i} \mathbf{W}^{\text{CNN}}(\mathbf{x}) \, \mathbf{x} = \mathbf{W}_i^{\text{Tokens}}(\mathbf{x}) \, \mathbf{x} .} \tag{9}$$

Here, $\mathbf{p}_i \in \mathbb{R}^D$ corresponds to the $i$th column in the token matrix $\mathbf{P}$, $\mathcal{T}_i$ extracts the respective features from the CNNs' output with $\mathbf{W}^{\mathcal{T}_i}$ denoting the corresponding linear matrix. $\mathbf{W}^{\text{CNN}}$ is the dynamic linear mapping computed by the B-cos CNN, and as color-indicated, $\mathbf{W}_i^{\text{Tokens}}(\mathbf{x}) = \mathbf{W}^{\mathcal{T}_i} \mathbf{W}^{\text{CNN}}(\mathbf{x})$.

Finally, note that we did not include the additive positional embedding $\mathbf{E}$, cf. Eq. (2) (left). For a detailed discussion on how to provide positional information to the B-cos ViTs, please see Sec. 3.5.

## 3.3 B-cos Attention

Interestingly, the attention operation itself is already dynamic linear and, as such, attention lends itself well to be integrated into the linear model summary according to Eq. (6). To ensure that the resulting linear transformation maintains the desired interpretability, we discuss the necessary changes to make the attention layers compatible with the B-cos formulation as discussed in Sec. 3.1.

**B-cos Attention.** Note that conventional attention indeed computes a dynamic linear transform:

$$\text{Attention}(\mathbf{P}; \mathbf{Q}, \mathbf{K}, \mathbf{V}) = \underbrace{\text{softmax}\left( \mathbf{P}^T \mathbf{Q}^T \mathbf{K} \mathbf{P} \right)}_{\text{Attention matrix } \mathbf{A}(\mathbf{P})} \underbrace{\mathbf{V} \mathbf{P}}_{\text{Value}(\mathbf{P})} = \underbrace{\mathbf{A}(\mathbf{P}) \, \mathbf{V}}_{\mathbf{W}(\mathbf{P})} \, \mathbf{P} = \mathbf{W}(\mathbf{P})\mathbf{P}. \tag{10}$$

Here, $\mathbf{Q}, \mathbf{K}$ and $\mathbf{V}$ denote the respective query, key, and value transformation matrices and $\mathbf{P}$ denotes the input tokens to the attention layer; further, softmax is computed column-wise.

In multi-head self-attention (MSA), see Eq. (3) (left), $H$ distinct attention heads are used in parallel after normalising the input. Their concatenated outputs are then linearly projected by a matrix $\mathbf{U}$:

$$\text{MSA}(\widetilde{\mathbf{P}}) = \mathbf{U} \left[ \mathbf{W}_1(\widetilde{\mathbf{P}})\widetilde{\mathbf{P}}, \ \mathbf{W}_2(\widetilde{\mathbf{P}})\widetilde{\mathbf{P}}, \ ... \ , \mathbf{W}_H(\widetilde{\mathbf{P}})\widetilde{\mathbf{P}} \right] \quad \text{with} \quad \widetilde{\mathbf{P}} = \text{LayerNorm}(\mathbf{P}) \tag{11}$$

While this can still[2] be expressed as a dynamic linear transform of $\mathbf{P}$, we observe the following issues with respect to the requirements (a-c), see Sec. 3.1. First, (a) while the attention matrix $\mathbf{A}(\mathbf{P})$ is bounded, the value computation $\mathbf{VP}$ and the projection by $\mathbf{U}$ are not. Therefore, as the output can arbitrarily increased by scaling $\mathbf{V}$ and $\mathbf{U}$, (b) a high weight alignment is not necessary to obtain large outputs. Finally, by normalising the outputs of the previous layer, the scale of those outputs does not affect the scale of the overall model output anymore, which violates requirement (c).

To address (a+b), we propose to formulate a B-cos Attention Block as follows. First, we replace the value computation and the linear projection by $\mathbf{U}$ by corresponding B-cos transforms. As in Böhle et al. (2022), we employ MaxOut and for a given input $\mathbf{P}$ the resulting projections are computed as

$$\text{B-cos Linear}(\mathbf{P}; \mathbf{S}) = \text{MaxOut} \circ \text{B-cos}(\mathbf{P}; \mathbf{S}) = \mathbf{W^S}(\mathbf{P})\mathbf{P} \quad \text{with} \quad \mathbf{S} \in \{\mathbf{U}, \mathbf{V}\}. \tag{12}$$

To fulfill (c), whilst not foregoing the benefits of LayerNorm[3], we propose to exclusively apply LayerNorm before the computation of the attention matrix. i.e. we compute $\mathbf{A}(\mathbf{P})$, see Eq. (10) as

$$\mathbf{A}(\mathbf{P}; \mathbf{Q}, \mathbf{K}) = \text{softmax}\left(\widetilde{\mathbf{P}}^T \mathbf{Q}^T \mathbf{K} \widetilde{\mathbf{P}}\right) \quad \text{with} \quad \widetilde{\mathbf{P}} = \text{LayerNorm}(\mathbf{P}). \tag{13}$$

In total, a 'B-cos AttBlock' thus computes the following linear transformation:

$$\text{B-cos AttBlock}(\mathbf{P}) = \left(\mathbf{W^U}(\mathbf{P}') \left[\mathbf{A}_h(\mathbf{P})\mathbf{W}_h^{\mathbf{V}}(\mathbf{P})\right]_{h=1}^{H} + \mathbf{I}\right)\mathbf{P} = \mathbf{W}^{\text{Att}}(\mathbf{P})\,\mathbf{P}, \tag{14}$$

Here, $\mathbf{P}' = \left[\mathbf{A}_h(\mathbf{P})\mathbf{W}_h^{\mathbf{V}}(\mathbf{P})\right]_{h=1}^{H}$, $\mathbf{W}^U$ and $\mathbf{W}^V$ as in Eq. (12), and $\mathbf{A}_h(\mathbf{P})$ as in Eq. (13); the identity matrix $\mathbf{I}$ reflects the skip connection around the MSA computation, see Fig. 2 and Eq. (3).

For an ablation study regarding the proposed changes, we kindly refer the reader to the supplement.

## 3.4 INTERPRETABLE MLPs AND CLASSIFIERS

To obtain dynamic linear and interpretable MLPs, we convert them to 'B-cos' MLPs, such that they are compatible with the B-cos formulation (Sec. 3.1) and align their weights with relevant inputs.

**B-cos MLPs.** Typically, an MLP block in a ViT computes the following:

$$\text{MLPBlock}(\mathbf{P}) = \text{Linear}_2 \circ \text{GELU} \circ \text{Linear}_1 \circ \text{LayerNorm}(\mathbf{P}) + \mathbf{P} \tag{15}$$

Here, Linear is as in Eq. (5), the GELU activation function is as in Hendrycks & Gimpel (2016), and LayerNorm as in Ba et al. (2016).

To obtain our 'B-cos MLPBlock', we follow Böhle et al. (2022) and replace the linear layers by B-cos transforms, remove the non-linearities and the normalisation (cf. Sec. 3.3); further, each 'neuron' is modelled by two units, to which we apply MaxOut (Goodfellow et al., 2013). As a result, each MLP block becomes dynamic linear:

$$\text{B-cos MLPBlock}(\mathbf{P}) = (\mathbf{M}_2(\mathbf{P})\,\mathbf{L}_2(\mathbf{P})\,\mathbf{M}_1(\mathbf{P})\,\mathbf{L}_1(\mathbf{P}) + \mathbf{I})\mathbf{P} = \mathbf{W}^{\text{MLP}}(\mathbf{P})\,\mathbf{P}. \tag{16}$$

Here, $\mathbf{M}_i(\mathbf{P})$ and $\mathbf{L}_i(\mathbf{P})$ correspond to the effective linear transforms performed by the MaxOut and B-cos operation respectively, and $\mathbf{I}$ denotes the identity matrix stemming from the skip connection.

**Classifier.** Similar to the B-cos MLPs, we also replace the linear layer in the classifier, cf. Eq. (5) (left), by a corresponding B-cos transform and the B-cos Classifier is thus defined as

$$\text{B-cos Classifier}(\mathbf{P}) = \text{B-cos} \circ \text{Pool}(\mathbf{P}) = \mathbf{L}(\mathbf{P}')\mathbf{W}^{\text{AvgPool}}\mathbf{P} = \mathbf{W}^{\text{Class}}(\mathbf{P})\,\mathbf{P}, \tag{17}$$

with $\mathbf{L}(\mathbf{P}')$ the dynamic linear matrix corresponding to the B-cos transform and $\mathbf{P}' = \text{Pool}(\mathbf{P})$.

## 3.5 POSITIONAL INFORMATION IN B-COS TRANSFORMERS

In contrast to CNNs, which possess a strong inductive bias w.r.t. spatial relations (local connectivity), transformers (Vaswani et al., 2017) are invariant w.r.t. the token order and thus lack such a 'locality

---

[2]To be exact, it can be represented as a dynamic *affine* transform, since LayerNorm adds a bias term.

[3]We noticed normalised inputs to be crucial for the computation of the attention matrix $\mathbf{A}(\mathbf{P})$ in Eq. (10): for unconstrained inputs, softmax easily saturates and suffers from the vanishing gradient problem.

bias'. To nevertheless leverage spatial information, it is common practice to break the symmetry between tokens by adding a (learnt) embedding $\mathbf{E}$ to the input tokens $\mathbf{P}$, see Eq. (2) (left).

However, within the B-cos framework, this strategy is not optimal: in particular, note that each B-cos transformation needs to align its weights with its inputs to forward a large output to the next layer, see Eq. (8) and Böhle et al. (2022). As a result, a B-cos ViT would need to associate contents (inputs) with specific positions, which could negatively impact the model's generalisation capabilities.

Therefore, we investigate two alternative strategies for providing positional information to the B-cos ViTs: additive and multiplicative attention priors, see Eqs. (18) and (19) respectively. Specifically, we propose to add a learnable bias matrix $\mathbf{B}_h^l$ to each attention head $h$ in every layer $l$ in the model. This pair-wise (between tokens) bias is then either added[4] before the softmax operation or multiplied to the output of the softmax operation in the following way (omitting sub/superscripts):

$$\mathbf{A}_{\text{add}}(\mathbf{P}) = \text{SM}\left(\mathbf{R}(\mathbf{P}) + \mathbf{B}\right) \quad (18) \qquad \text{and} \quad \mathbf{A}_{\text{mul}}(\mathbf{P}) = \text{SM}\left(\mathbf{R}(\mathbf{P})\right) \times \text{SM}\left(\mathbf{B}\right) \ . \quad (19)$$

Here, $\mathbf{R}(\mathbf{P}) = \mathbf{Q}\widetilde{\mathbf{P}}\widetilde{\mathbf{P}}^T\mathbf{K}^T$ and SM denotes softmax. The bias $\mathbf{B}$ thus allows the model to learn an *attention prior*, and the attention operation is no longer invariant to the token order. As such, the model can learn spatial relations between tokens and encode them explicitly in the bias matrix $\mathbf{B}$. In our experiments, this significantly improved the performance of the B-cos ViTs, see Sec. 5.1.

## 4 EXPERIMENTAL SETTING

**Dataset.** In this work, we focus on Vision Transformers (ViTs, Dosovitskiy et al. (2021)) for image classification. For this, we evaluate the B-cos and conventional ViTs and their explanations on the ImageNet dataset (Deng et al., 2009). We use images of size 224×224. For B-cos models, we encode the images as in Böhle et al. (2022).

**Models.** We follow prior work and evaluate ViTs of different sizes in common configurations: Tiny (Ti), Small (S), and Base (B), cf. Steiner et al. (2021). We train these models on the frozen features of publicly available (Böhle et al., 2022; Marcel & Rodriguez) (B-cos) DenseNet-121 models and extract those features at different depths of the models: after 13, 38, or 87 layers. Model names are thus as follows: (B-cos) ViT-{size}-{L} with size∈{Ti, S, B} and L∈{13, 38, 87}. We opted for (B-cos) DenseNet-121 backbones, as the conventional and the B-cos version achieve the same top-1 accuracy on the ImageNet validation set. In particular, we compare B-cos ViTs on B-cos backbones to normal ViTs on normal backbones.

**Training.** We employ a simple training paradigm that is common across models for comparability. All models are trained with RandAugment (Cubuk et al., 2020) for 100 epochs with a learning rate of $2.5e^{-4}$, which is decreased by a factor of 10 after 60 epochs; for details, see supplement.

**Evaluation Metrics.** We evaluate all models with respect to their accuracy on the ImageNet validation set. Further, we employ two common metrics to assess the quality of the model explanations.

First, we evaluate the grid pointing game (Böhle et al., 2021). For this, we evaluate the explanations (see below) on 250 synthetic image grids of size 448×448, containing 4 images of distinct classes, see Fig. 4; the individual images are ordered by confidence and we measure the fraction of positive attribution an explanation method assigns to the correct sub-image when explaining a given class.

Note that, in contrast to fully convolutional networks, transformers with positional embeddings expect a fixed-size input. To nevertheless evaluate the models on such synthetic image grids, we scale down the image grid to the required input size of 224x224 to allow for applying the ViTs seamlessly.

Second, we evaluate two pixel perturbation metrics, cf. Chefer et al. (2021). For this, the pixels are ranked according to the importance assigned by a given explanation method. Then, we increasingly zero out up to 25% of the pixels in increasing (decreasing) order, whilst measuring the model confidence in the ground truth class; a good explanation should obtain a high area under (over) the curve, i.e. the model should be insensitive to unimportant pixels and sensitive to important ones.

We evaluate the perturbation metrics on the 250 most confidently and correctly classified images to enable a fair comparison between models, as the confidence affects the metrics; more details in supplement. Last, to succinctly summarise the two metrics, we evaluate the area *between* the curves.

---

[4]Note that an additive positional bias in attention layers has been proposed before (Graham et al., 2021).

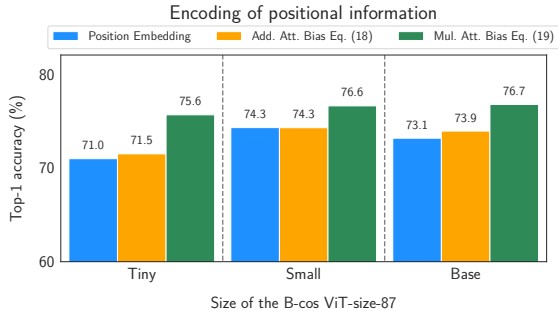

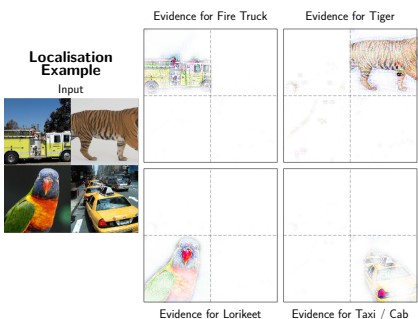

Fig. 3: ImageNet accuracy of differently sized B-cos ViTs (Tiny, Small, Base) depending on the positional encoding. We find B-cos ViTs with $\mathbf{A}_{\mathrm{mul}}$, see Eq. (19), to perform significantly better.

Fig. 4: In the localisation metric, we measure the fraction of pos. evidence assigned to the correct grid cell for each occurring class.

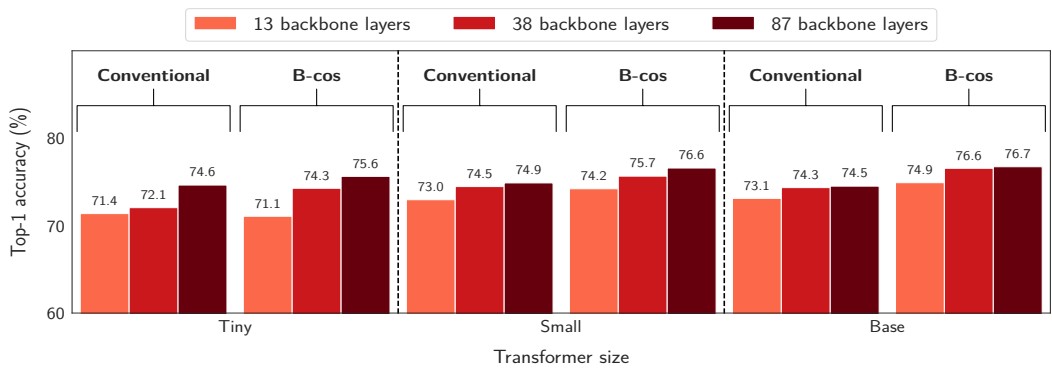

Fig. 5: ImageNet accuracies of B-cos ViTs with a multiplicative attention bias (Eq. (19)) compared to standard ViTs and backbones, both for differently sized ViTs (Tiny, Small, Base) and backbones (13, 38, or 87 layers). We find that the B-cos ViTs perform at least as well as the baseline ViTs over almost all tested configurations.

**Explanation Methods.** Apart from the model-inherent explanations (Eq. (7)), we evaluate two sets of explanation methods. First, we follow Chefer et al. (2021) and evaluate common *transformer-specific* explanations such as the attention in the final layer (FinAtt), attention rollout (Rollout) (Abnar & Zuidema, 2020), a transformer-specific LRP implementation (CheferLRP) proposed by Chefer et al. (2021), 'partial LRP'(pLRP) (Voita et al., 2019), and 'GradSAM' (Barkan et al., 2021). Further, we evaluate *architecture-agnostic* methods such as Integrated Gradients (IntGrad) (Sundararajan et al., 2017), adapted GradCAM (Selvaraju et al., 2017) as in Chefer et al. (2021), and 'Input×Gradient' (IxG), cf. Adebayo et al. (2018). As no LRP rules are defined for B-cos ViTs we only apply it to baseline models. For method details, we kindly refer the reader to the supplement.

We evaluate all of those methods (if applicable) to the proposed B-cos ViTs, as well as the baselines consisting of conventional ViTs and backbones and compare them on the metrics described above.

## 5 RESULTS

In the following, we present our experimental results. Specifically, in Sec. 5.1 we analyse the classification performance of the B-cos ViTs: we investigate how the encoding of positional information affects model accuracy (see Sec. 3.5) and compare the classification performance of B-cos and conventional ViTs. Further, in Sec. 5.2, we evaluate the model-inherent explanations of the B-cos ViTs against common post-hoc explanation methods evaluated on the same models. To highlight the *gain in interpretability* over conventional ViT models, we also compare the inherent explanations of the B-cos ViTs to the best post-hoc explanations evaluated on conventional ViTs, see supplement.

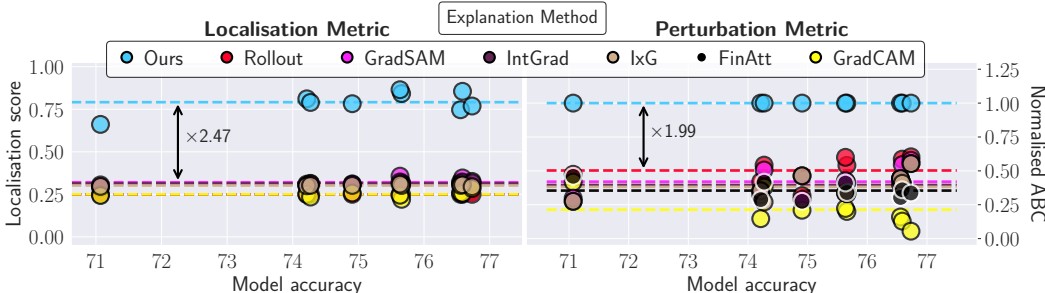

Fig. 6: Quantitative comparison of explanation methods according to two metrics: localisation (**left**) and perturbation (**right**); for a description of metrics and methods, see Sec. 4. We evaluated the methods for all B-cos ViTs shown in Fig. 5 and plot the corresponding scores (markers). We also plot the mean score over all models (dashed lines) per method and the average improvement of the *model-inherent* over the best *post-hoc* explanation (localisation: ×2.47, perturbation: ×1.99). Note that for the perturbation metric, we normalised the area between curves (ABC) by the scores of the model-inherent explanations for better cross-model comparison.

## 5.1 CLASSIFICATION PERFORMANCE OF B-COS VITs

In Fig. 3, we compare the top-1 ImageNet accuracy of various B-cos ViTs trained on the feature embeddings of the 87th layer of a frozen[5] B-cos DenseNet-121 (Böhle et al., 2022). Specifically, we compare ViTs of different sizes (Tiny, Small, Base) and with different ways of allowing the models to use positional information, see Eqs. (2), (18) and (19). We find that the multiplicative attention bias, see Eq. (19), consistently yields significant gains in performance. As discussed in Sec. 3.5, we believe this could be due to the higher disentanglement between content and positional information. However, in preliminary experiments with *conventional* ViTs, we did not observe significant benefits from such a multiplicative prior and this seems to be particularly advantageous for B-cos ViTs.

Interestingly, once trained with such a multiplicative attention prior, we find the B-cos ViTs to perform at least as good as their conventional counterparts over a wide range of configurations, see Fig. 5; we find consistent results even without MaxOut in the Transformer layers (cf. Sec. 3), as we show in Appendix B.3. However, these results have to be interpreted with caution: ViTs are known to be highly sensitive to, e.g., the amount of data augmentation, the number of training iterations, and model regularisation, see Steiner et al. (2021). Moreover, our goal in this work is to develop interpretable ViTs and our focus thus lies on evaluating the quality of the explanations (Sec. 5.2).

## 5.2 INTERPRETABILITY OF B-COS VITs

Here, we assess how well the inherent explanations (Eq. (7)) of B-cos ViTs explain their output and compare to common post-hoc explanations; for comparisons to baseline ViTs, see supplement.

**Localisation Metric.** In Fig. 6 (left), we plot the mean localisation score per model configuration (B-cos ViT-{size}-{L}) and explanation method, see Sec. 4. We find that across all configurations, the model-inherent explanations according to Eq. (7) yield by far the best results under this metric and outperform the best *post-hoc* explanation for the B-cos ViTs (Rollout) by a factor of 2.47.

**Pixel Perturbation.** As for the localisation, in Fig. 6 (right), we plot the normalised mean area between the curves (ABC) per model configuration and explanation method of the B-cos ViTs. Specifically, the mean ABC is computed as the mean area between the curves when first removing the most / least important pixels from the images; we normalise the mean ABC for each explanation by the mean ABC of the model-inherent explanation (Ours) per model configuration to facilitate cross-model comparisons. Again, the model-inherent explanations perform best and, on average, they outperform the second best post-hoc method (Rollout) on B-cos ViTs by a factor of 1.99.

**Qualitative Examples.** In Figs. 1 and 7, we qualitatively compare the inherent explanations (size: B, 38 backbone layers, see Fig. 5) to post-hoc explanations evaluated on the same model. As becomes apparent, the model-inherent summaries not only perform well quantitatively (cf. Fig. 6), but are also qualitatively convincing. Colour visualisations as in Böhle et al. (2022); more results in supplement. In contrast to attention explanations, which are not class-specific (Chefer et al., 2021), we find the

---

[5]We chose to freeze the backbones to reduce the computational cost and compare the architectures across a wide range of settings. We observed comparable results when training the full models for individual architectures.

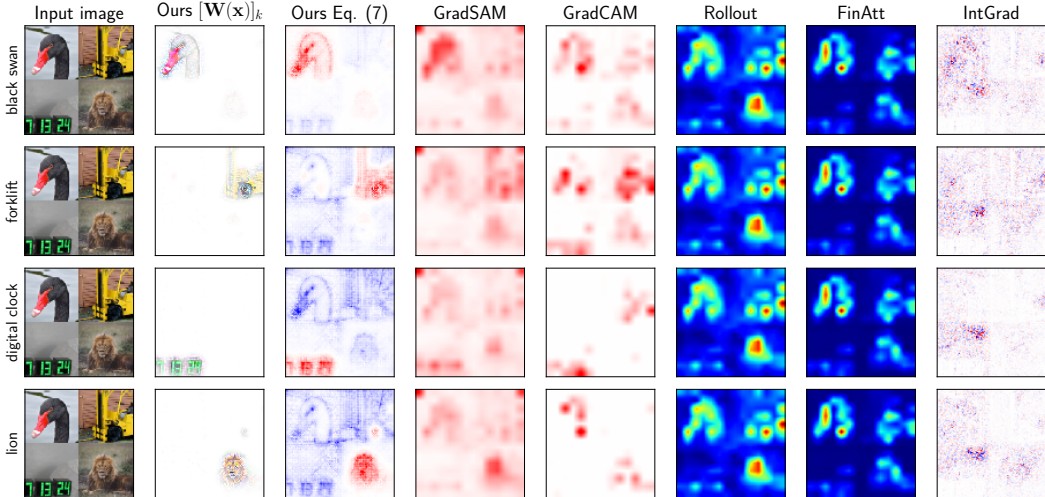

Fig. 7: Comparison of the model-inherent explanations (Ours) of a B-cos ViT-B-38, and several post-hoc explanations (GradSAM, GradCAM, Rollout, FinAtt, IntGrad) for class $k$ (left). In particular, we show explanations for the classes 'black swan', 'forklift', 'digital clock', and 'lion' on a synthetic image containing these classes, as used in the localisation metric, see Fig. 4. As B-cos ViTs follow the B-cos formulation, we can visualise the rows of $\mathbf{W}(\mathbf{x})$ in colour (Böhle et al., 2022). Additionally, we show contribution maps according to Eq. (7).

model-inherent explanations of B-cos ViTs to be highly detailed and class-specific. E.g., in Fig. 1, we compare model-inherent explanations to attention-based explanations for single images from the ImageNet dataset which are inherently ambiguous. In Fig. 7, we evaluate the model on images as used in the localisation metric, see Sec. 4, i.e. synthetic images with multiple classes. In both cases we find the model-inherent explanations to accurately highlight the respective features for the class logit that we aim to explain, whereas other methods are much less sensitive to the class logit; in fact, attention-based explanations are inherently agnostic to the choice of logit and thus the same for all classes. For comparisons to explanations for conventional ViTs, see the supplement.

## 6  CONCLUSION

We present a novel approach for designing ViTs that are *holistically* explainable. For this, we design every component of the ViTs with the explicit goal of being able to summarise the *entire model* by a single linear transform. By integrating recent advances in designing interpretable dynamic linear models (Böhle et al., 2022), these summaries become interpretable, as they are implicitly optimised to align with relevant input patterns. The resulting B-cos ViTs constitute competitive classifiers and their inherent linear summaries outperform any post-hoc explanation method on common metrics.

Compared to attention-based explanations, our method can be understood to 'fill the blanks' in attention rollout (Abnar & Zuidema, 2020). Specifically, attention rollout computes a linear summary of the attention layers only. By integrating explanations for the remaining components (tokenisation, attention, MLPs), we are able to obtain *holistic* explanations of high detail, see Figs. 1 and 7.

As transformers are highly modality-agnostic, we believe that our work has the potential to positively impact model interpretability across a wide range of domains. Evaluating B-cos transformers on different tasks and modalities is thus an exciting direction that we aim to explore in future work.

**Limitations.** While the B-cos ViTs allow us to extract model-faithful explanations for single images, note that these explanations are always *local* in nature, i.e. for single data points. The explanations thus help understanding an *individual* classification, but do not directly give insights into which features the models most focus on over the *entire dataset*. It would thus be interesting to combine B-cos ViTs with *global* explanation methods, such as in Bau et al. (2017); Kim et al. (2018).

Further, we focused primarily on the designing of interpretable transformers, and, to test across a wide range of models, limited experiments to the ImageNet-1k dataset. As transformers are known to significantly benefit from additional data and training (Dosovitskiy et al., 2021), it would be interesting to test the limits of capacity of the B-cos ViTs and scale to more complex tasks.

ETHICS STATEMENT

The growing adoption of deep neural network models in many different settings is accompanied by an increasingly louder call for more transparency in the model predictions; especially in high-stake situations, relying on an opaque decision process can have severe consequences (Rudin, 2019). With this work, we make a step towards developing inherently more transparent neural network models that explain their decisions without incurring losses in model performance.

However, we would like to emphasise that our contribution can only be seen as a step in this direction; while the explanations might seem meaningful on a per sample basis, they could lead to a false sense of security in terms of 'understanding' model behaviour. Currently, we can give no formal guarantees for model behaviour under unseen input data and more research on explainable machine learning is necessary. Lastly, any research that holds the potential for accelerating the adoption of machine learning systems could have unpredictable societal impacts.

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

# Supplementary Material

## Table of Contents

In this supplement to our work on designing holistically explainable transformers, we provide:

# A ADDITIONAL QUALITATIVE RESULTS

> In this section, we show additional *qualitative* results and discuss the qualitative differences between the various explanations methods in more detail. For this, we include explanations for B-cos as well as for conventional ViTs.

> In this section, we describe the model architectures, the training procedure, the explanation methods, as well as the evaluation metrics in more detail.

## A.1 COMPARISON TO ATTENTION EXPLANATIONS

For the reader's convenience, in Fig. A1 we repeat the qualitative results presented in Fig. 1, such as to facilitate the following discussion of the qualitative differences between the holistic and the purely attention-based explanations of B-cos ViTs.

In particular, we would like to point out several key differences between our *holistic* explanations as per Eq. (7) and the attention-based explanations according to Attention Rollout (Abnar & Zuidema, 2020) and the last layer's attention.

First, only the linear mapping $\mathbf{W}(\mathbf{x})$ used for the contribution maps in Eq. (7) is able to capture 'negative evidence' for the respective classes, see col. 2 in Fig. A1. Note that this does not depend on the particular choice of images shown here. Instead, as the attention matrices consist only of non-negative values, the attention-based explanations cannot distinguish between *positively* and *negatively* contributing features. Therefore, to improve the attention visualisation and more clearly highlight details in the attention maps, we plot the attention-based explanations on a colour scale from 0 to the maximal attention value. The model-inherent explanation,

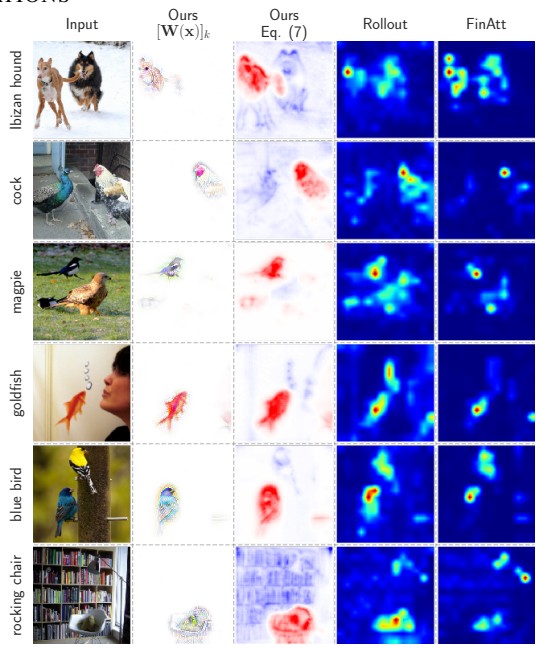

Fig. A1: Inherent explanations (cols. 2+3) of B-cos ViTs vs. attention explanations (cols. 4+5) for the same model. Note that $\mathbf{W}(\mathbf{x})$ faithfully reflects the *whole* model and yields more detailed and class-specific explanations than attention alone. For a detailed discussion, see Sec. A.1.

in contrast, use a colour scale from $[-p, p]$ with $p$ the maximum absolute pixel contribution.

Importantly, as attention-based explanations do not distinguish between positively and negatively contributing neurons / pixels, they are inherently not *class-specific*. For example, especially in images in which various classes are present (rows 1, 2, 3, 5), attention focuses on all occurring class instances: all birds in rows 2, 3, and 4, as well as both dogs in row 1. The model-inherent explanations, on the other hand, clearly distinguish between positive and negative contributions and are thus able to resolve class-specific details.

Secondly, the attention-based explanation are of much lower resolution than the model-inherent ones. Again, this cannot be attributed to the choice of images, but reflects an intrinsic difference between the explanations: whereas the linear mapping $\mathbf{W}(\mathbf{x})$ reflects the entire model including the tokenisation module and thus attributes on the level of *pixels*, the attention explanations can only yield attributions at the level of *tokens*, which highlights a key difference between the methods: while the attention explanations only include a few layers in their attributions, the model-inherent linear map $\mathbf{W}(\mathbf{x})$ constitutes an exact summary of the *entire model*.

Third, as shown in the first column, the rows of the linear mapping $\mathbf{W}(\mathbf{x})$ can directly be visualised in colour space, as the B-cos ViTs are designed according to the B-cos framework proposed by Böhle et al. (2022). Note that this is not a masked version of the original image, but instead a direct reflection of the dynamically computed weight matrix $\mathbf{W}(\mathbf{x})$, for details see Sec. C.3. Such visualisations are not possible with attention-based explanations.

Finally, we would like to highlight the relation between attention rollout and the model-inherent linear mapping. Specifically, as shown in Eq. (6), note that the entire model can be summarised by

$$\mathbf{W}(\mathbf{x}) = \mathbf{W}^{\text{Class}}(\mathbf{x}) \ \prod_{l=1}^{L} \left( \mathbf{W}_l^{\text{MLP}}(\mathbf{x}) \ \mathbf{W}_l^{\text{Att}}(\mathbf{x}) \right) \ \mathbf{W}^{\text{Tokens}}(\mathbf{x}) \,. \tag{A.1}$$

Interestingly, attention rollout in fact computes the overall attention attributions in a similar manner:

$$\mathbf{A}_{\text{rollout}}(\mathbf{x}) = \mathbf{I}^{\text{Class}} \ \prod_{l=1}^{L} \left( \mathbf{I}_l^{\text{MLP}} \ \bar{\mathbf{A}}_l(\mathbf{x}) \right) \ \mathbf{I}^{\text{Tokens}} \,, \tag{A.2}$$

with $\mathbf{I}^{\text{layer}}$ replacing the actual linear transformation of a specific layer by an identity matrix and $\bar{\mathbf{A}}_l$ denoting the average attention distribution of layer $l$, see Abnar & Zuidema (2020). Comparing Eqs. (A.1) and (A.2) succinctly shows how solely attention-based explanations leave out a large part of the model computations, which are seamlessly integrated in the complete linear mapping given by $\mathbf{W}(\mathbf{x})$.

### A.2    NORMALISING THE VISUALISATIONS ACROSS MULTIPLE EXPLANATIONS

As we discuss in Appendix C.3, the individual explanations are normalised independently of each other; i.e., all explanations shown on the blue-white-red colour map (Ours (Eq. (7)), GradSAM, GradCAM, IxG, IntGrad, and pLRP) are plotted on a scale from $-v$ to $v$ with $v$ the 99.9th percentile of the absolute value of the given attribution map. As such, the resulting attribution maps are not directly comparable across different explanations. To show the effect of normalising across multiple explanations, in Fig. A2 we repeat Fig. 7 from the main paper, once normalised *across* contribution maps, and once normalised independently. To normalise across contribution maps, $v$ is computed as the 99.9th percentile of the absolute value in all of the four class explanations per method in the figure.

### A.3    ADDITIONAL EXPLANATIONS AND COMPARISONS

In Fig. A3, we compare the model-inherent explanations of a B-cos ViT-B-38 model to additional explanation methods apart from purely attention-based ones. Moreover, we show explanations extracted for a *conventional* ViT-B-38 model in Fig. A4 for comparison.

We would like to highlight the following. First, we find that the model-inherent explanations of the B-cos ViTs provide more detailed and convincing explanations than any of the post-hoc explanations when evaluated on the same model. Crucially, these explanations do not only look convincing, but in fact accurately reflect the model computations of the B-cos ViTs—i.e., they are *model-faithful*.

Further, the model-inherent explanations do not only compare favourably to other explanations evaluated on our newly proposed B-cos ViTs. Instead, they also provide much more detail and highlight more class-specific features than any of the post-hoc explanation methods yield on conventional ViTs, see Fig. A4. As such, we find that there is a clear *gain in interpretability* when using B-cos ViTs instead of conventional ones.

## B    ADDITIONAL QUANTITATIVE RESULTS

In this section, we quantitatively compare the interpretability of the B-cos ViTs to that of conventional ViTs (Appendix B.1). Specifically, as in Sec. 5.2, we discuss the localisation and the perturbation metrics. Additionally, in Appendix B.2, we present results of an ablation study in which we investigate the design choices of the B-cos Attention module in more detail.

### B.1    INTERPRETABILITY COMPARISON: B-COS VS. CONVENTIONAL VIT MODELS

**Localisation.** In Fig. B1 (right) we present the localisation results of post-hoc explanation methods evaluated on conventional ViTs; for comparison, we repeat the results of the B-cos ViTs (see Fig. 6) on the left. As becomes apparent, no post-hoc explanation method evaluated on conventional ViTs allows for localising the correct grid images (cf. Fig. 4) in the localisation metric to the same degree as is possible with the model-inherent explanations. Specifically, we find that the model-inherent explanations yield on average 2.32 times higher localisation scores across the various model configurations than the best post-hoc explanation method on conventional ViTs (IntGrad).

**Perturbation.** In Fig. B2 (right) we present the perturbation metric results of post-hoc explanation methods evaluated on conventional ViTs; for comparison, we repeat the results of the B-cos ViTs (see Fig. 6) on the left. Specifically, as discussed in the main paper, on the left we show the normalised mean area between the curces (ABC) for each model configuration (differently sized backbones and transformers), in which the ABC of each configuration is normalised by the ABC of the model-inherent explanations (Ours). To enable a comparison between the conventional and the B-cos ViTs, on the right we normalise by the best *post-hoc method* (FinAtt) and further multiply the resulting score by the ratio between the mean scores across configurations of FinAtt on conventional ViTs and the mean scores of Ours on B-cos ViTs (corresponding to the respective dashed lines *before* normalisation).

As discussed in the main paper, we find that the model inherent explanations of B-cos ViTs consistently yield the best pixel ranking for each of the configurations of the B-cos ViTs. Further, the

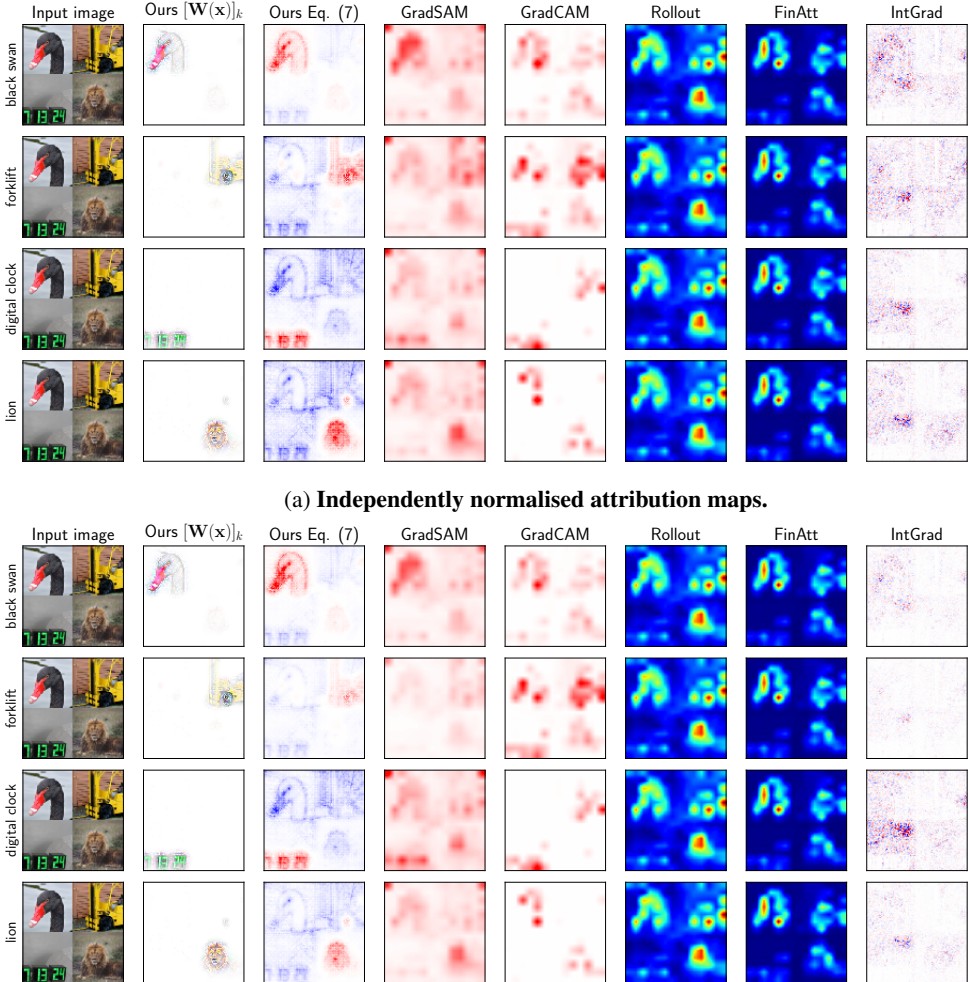

(a) **Independently normalised attribution maps.**

(b) **Jointly normalised attribution maps.**

Fig. A2: (a) Repetition of Fig. 7 from the main paper, i.e., with each attribution map normalised independently. (b) The same figure is shown again, but this time the normalisation is done *column-wise*, i.e., all explanations of a given method are shown on the same scale (except for the coloured explanations in the second column, which are unchanged). Note that the attention-based explanations are the same across all classes to begin with and are thus not affected. The IntGrad explanations as well as the forklift explanation according to Ours (7) change most notably—in the case of the model-inherent explanations, this directly reflects the fact that the model has found the least evidence for forklift: the sum of positive contributions according to the model-inherent contribution maps are 13.6 (black swan), 3.3 (forklift), 7.0 (digital clock), and 5.2 (lion).

ABC is on average much higher for the model-inherent explanations for B-cos ViTs than the ABC resulting from the rankings of post-hoc explanations on conventional ViTs. Note, however, that this could also reflect a difference in model stability and the comparisons across models have thus to be interpreted with care.

### B.2 ABLATION STUDY: ANALYSING THE DESIGN CHOICES IN THE B-COS ATTENTION MODULE

In this subsection, we analyse the impact of the proposed changes for the attention module in more detail. Specifically, we investigate the effect of changing the position of the LayerNorm module within the attention layer. Further, we discuss the effect of changing the model's value computation and projection layers to B-cos layers.

**Position of the LayerNorm module.** In Fig. B3, we present the classification performance results of various B-cos ViT-S models trained on embeddings extracted at different depths of the back-

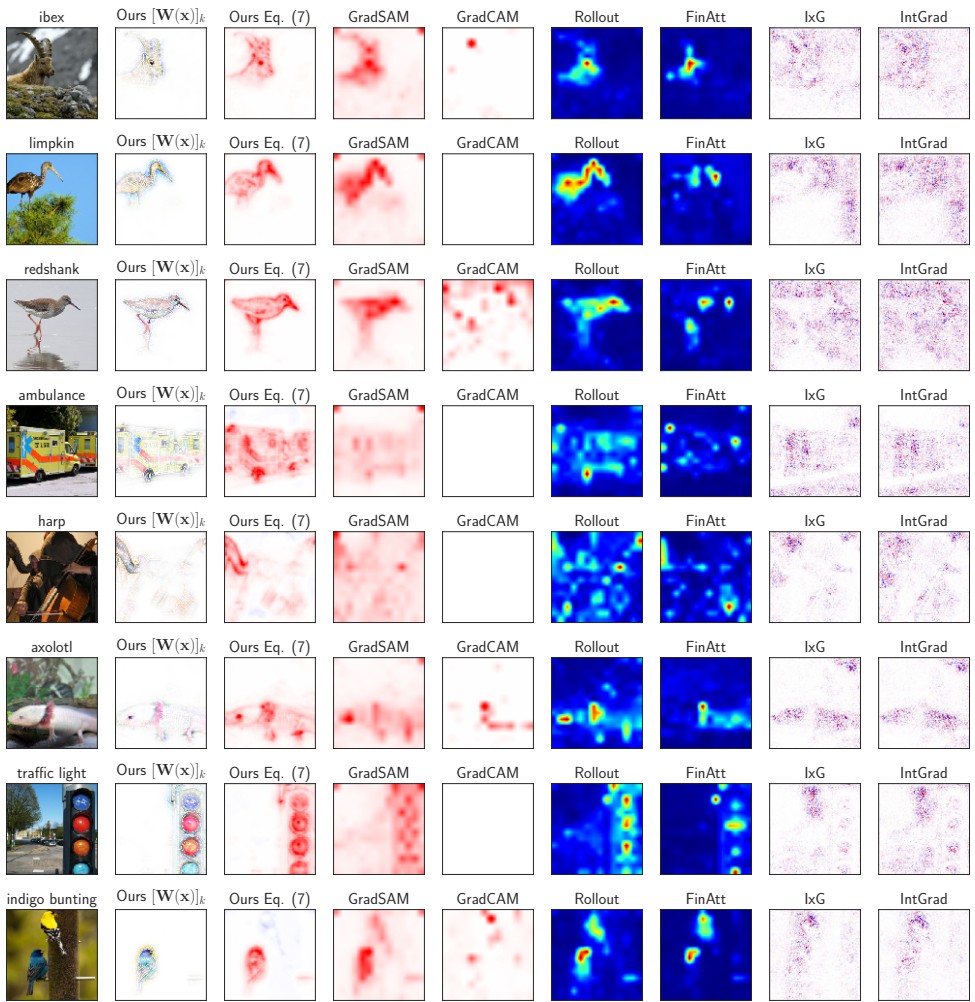

Fig. A3: Comparison of inherent explanations (Ours) of a B-cos ViT-B-38, and several post-hoc explanations for class $k$ (left). For Ours, we show a colour visualisation (see Sec. C.3) of the corresponding row of the weight matrix $\mathbf{W}(\mathbf{x})$ as well as the corresponding contribution maps as defined in Eq. (7). Note that of the compared methods, the model-inherent explanations yield by far the most detail and class-specificity, as also discussed in Sec. A.1 above. For a comparison to explanations generated for a conventional ViT-B-38 model on the same set of images, see the following Fig. A4.

bone B-cos DenseNet-121 model. Specifically, we evaluate three different model configurations for each backbone depth: 'Standard Attention', 'Shifted LayerNorm (B=1)', and 'Shifted LayerNorm (B=2)'. Here, 'Standard Attention' refers to the unchanged attention module as it is used in conventional ViT models. On the other hand, the 'Shifted LayerNorm' models implement the change described in eq. (13); i.e., for these models, the normalisation layer is moved inside the softmax computation instead of being applied before the attention module as a whole. We observe that the models with the shifted LayerNorm perform significantly better than those using standard attention, especially for shallower backbone models.

We attribute this to the fact that using LayerNorm only within the softmax computation leaves the norm of the value vectors unchanged, such that they are inherently on a similar scale as the token embeddings they are added to in the skip connection—this allows the model to compute significant and well-scaled updates of the token embeddings via the attention module. Moreover, the model can learn bias and scale parameters that are specifically tailored to the softmax layer, instead of affecting both the softmax computation as well as the value vectors at the same time.

In the standard attention module, on the other hand, the norm of the value vectors can differ significantly from the norm of the token embeddings in the residual connection, which can make it more

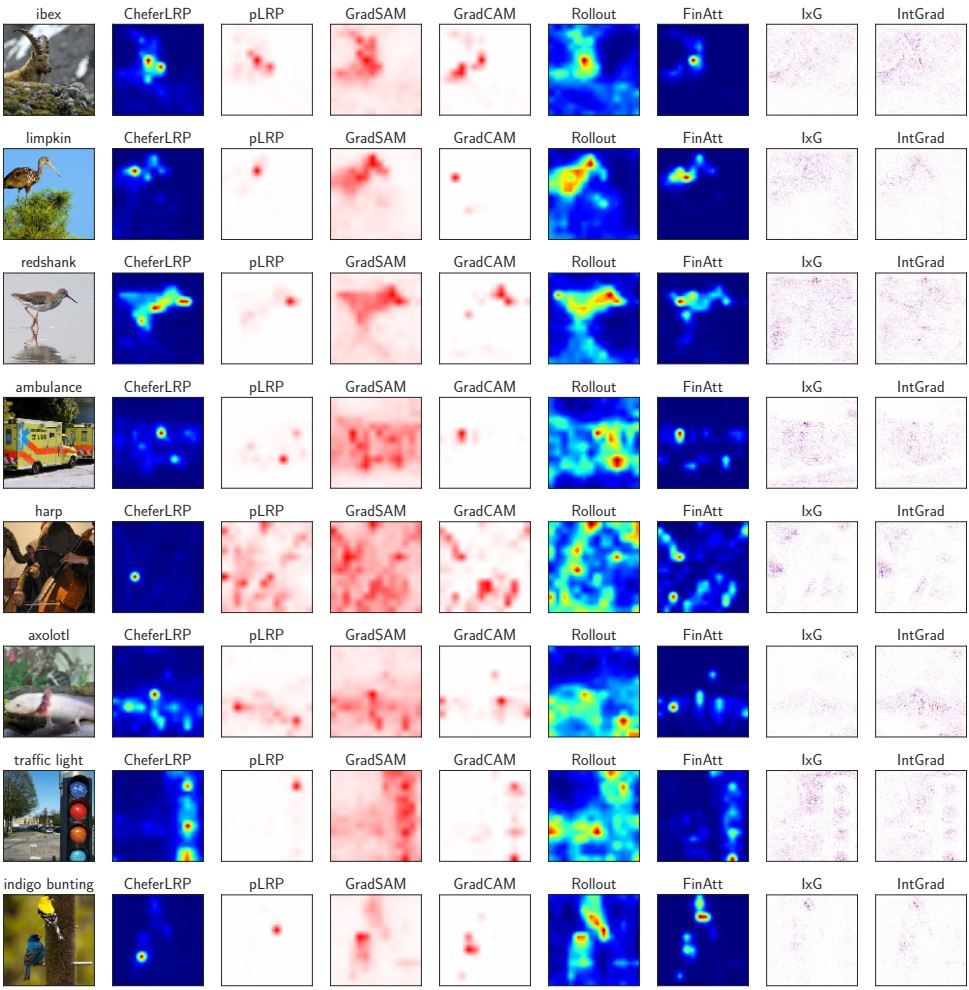

Fig. A4: Importance attributions given by common post-hoc explanation methods applied to a conventional ViT-B-38 model (see Sec. 4 for the model specifications) on the same set of images as in Fig. A3. We find that none of the common post-hoc explanations for conventional ViTs give similarly detailed results as the model-inherent explanations do for B-cos ViTs, see Fig. A3. As such, we observe a clear gain in interpretability when using B-cos ViTs instead of conventional ones.

difficult for the model to take advantage of the attention module. This hypothesis is corroborated by an analysis of the norms of the embeddings. In particular, we find the embeddings coming from the skip connections to be on average several orders of magnitude larger than those returned by the attention layer in the models trained with standard attention: e.g., by a factor of $10^5$ in the model trained on a 13-layer backbone model. In contrast, in the Shifted LayerNorm model with B=2, this factor is on the order of $10^0$, i.e., both embeddings are in fact on a similar scale. While models with standard attention could in principle learn large scale values in the LayerNorm modules, this would effectively result in one-hot encodings in the softmax computation, which could hamper learning.

As the attention modules thus have very limited impact on the model output, we further observe that the model does not seem to learn useful attention maps in the first place. E.g., in Fig. B4 we compare the attention maps of the models with Standard Attention and the Shifted LayerNorm models on various images and find those of the Standard Attention model to be much less structured; note that for the Standard Attention model, the model output is no longer a dynamic linear transformation of the input, due to the LayerNorm module.

**B-cos transforms for value and projection layers.** In the following, we discuss the impact of replacing the linear layers typically used in the value computation and the projection layers by B-cos layers (cf. eq. (14). In particular, we would first like to highlight that a B-cos transform with

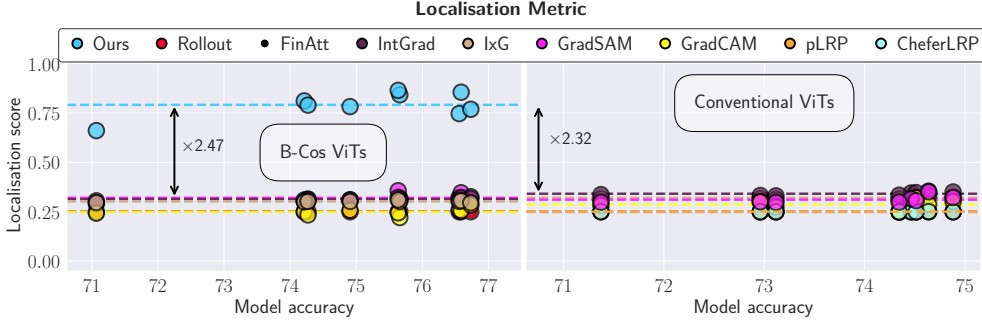

Fig. B1: **Left:** Localisation metric results for the B-cos ViTs of various sizes, same as shown in the main paper in Fig. 6 (left). **Right:** For comparison, we show the results of common post-hoc explanations evaluated on the corresponding conventional ViTs. As can be seen, the model-inherent explanations of the B-cos ViTs not only constitute the best localising explanation for any given B-cos ViT, but also achieve much higher localisation scores than the best post-hoc explanations evaluated on conventional ViTs.

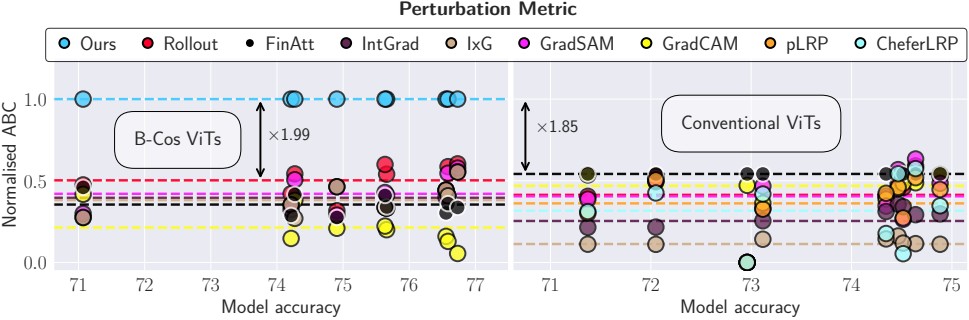

Fig. B2: **Left:** Perturbation metric results for the B-cos ViTs of various sizes, same as in the main paper in Fig. 6 (left). **Right:** For comparison, we show the results of common post-hoc explanations evaluated on the corresponding conventional ViTs. Note that, to allow for a comparison between the B-cos ViTs and the conventional ViTs, we normalised the mean ABCs of the conventional ViTs by the mean ABC of the best post-hoc method (FinAtt) and multiplied the results by the ratio between the mean ABCs *FinAtt* on conventional ViTs and the mean ABCs of *Ours* on B-cos ViTs. For a detailed discussion, see Sec. B.

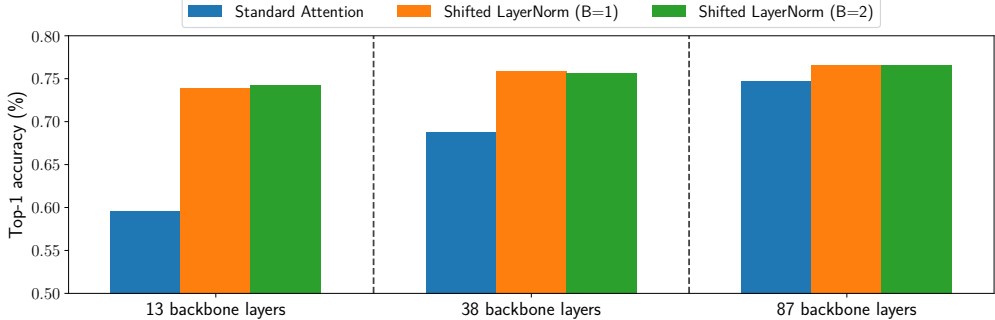

Fig. B3: Ablation results for different version of the attention module. Specifically, we show the top-1 accuracy on the ImageNet validation set for B-cos ViTs with the conventional attention module ('Standard Attention') and the proposed B-cos Attention ('Shifted LayerNorm', see eq. (13)). Specifically, for the latter we show the results of models trained with different values for B (B=1 and B=2) in the value computation and the projection head, see eq. (14). Note that for B=1, the B-cos transform is equivalent to a linear transformation. As such, 'Standard Attention' and 'Shifted LayerNorm (B=1)' differ only in the positioning of the LayerNorm module.

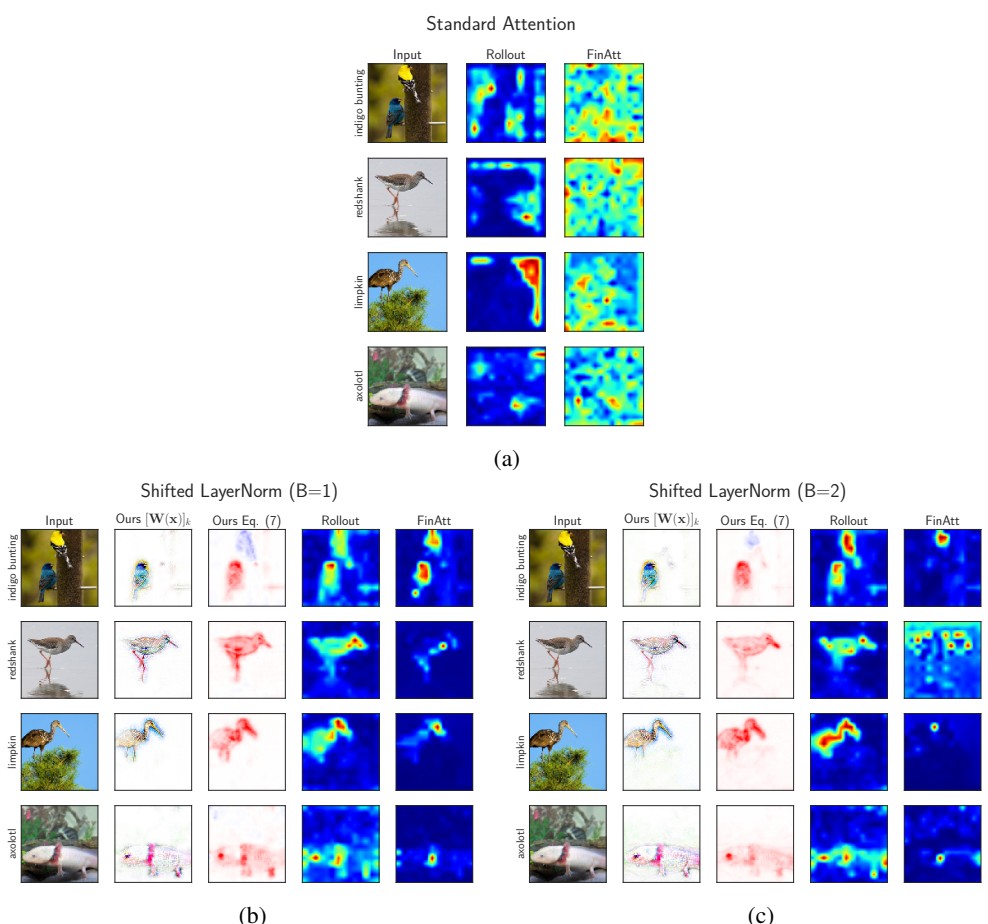

Fig. B4: Visualisation of attention-based explanations as well as the model-inherent explanations for models trained with **(a)** the standard attention module, **(b)** Shifted LayerNorm (B=1), and **(c)** Shifted LayerNorm (B=2); for details, see Sec. B.2. As we describe in that section, the models with standard attention seem unable to take advantage of the attention module and thus do not learn structured attention maps **(a)**. In contrast, once the LayerNorm is shifted inside the softmax computation, the models are not only inherently explainable by a single linear transformation, but also learn to use the attention layers in a much more structured manner **(b+c)**.

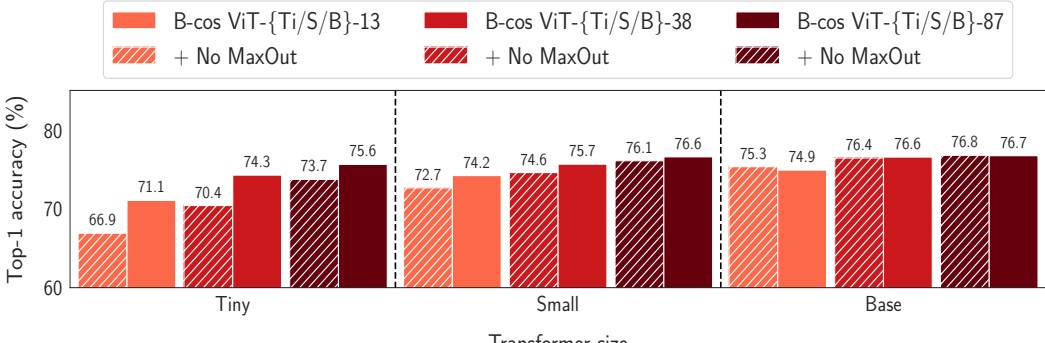

(a) Comparison between B-cos ViTs with (solid) and without (striped) MaxOut in the Transformer layers.

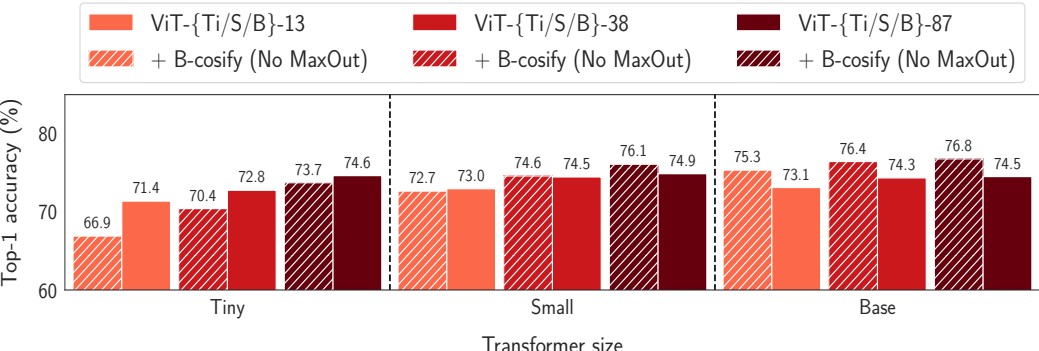

(b) Comparison between B-cos ViTs (no MaxOut) (striped) in the Transformer layers and conventional ViTs.

Fig. B5: **Understanding the impact of MaxOut on model performance.** **(a)** For the Tiny transformers, we find that MaxOut indeed significantly improves model performance. However, this performance gap between models with and without MaxOut closes with increasing model size (Small and Base). **(b)** As such, when comparing to conventional ViTs (same numbers as in Fig. 5), we find that B-cos ViTs without MaxOut can achieve similar performance without adding additional parameters in the Transformer layers, as long as the initial model size is sufficiently large.

B=1 is in fact equivalent to a linear layer and the Shifted LayerNorm (B=1) model thus only differs from the conventional attention by the placement of the LayerNorm module.

When comparing the classification performance of the Shited LayerNorm models with different values for B, we find the models to perform very similarly, see Fig. B3. Given that the main difference between those models is a slightly higher value of B in one of them, this is not surprising. Moreover, given that both models still adhere to the B-cos formulation, both models are accurately summarised by a global linear transformation and allow for a model-faithful decomposition into individual input contributions, see Fig. B4; note that all other modules still use B=2 in both models and thus already induce significant alignment, irrespective of the value and projection layers[6]. In contrast to the model with Standard Attention, we find both models with Shifted LayerNorm to learn highly structured attention maps, see Fig. B4.

### B.3 Ablation study: Analysing the impact of MaxOut on performance

As discussed in Sec. 3, when converting the baseline ViTs to B-cos ViTs, we follow Böhle et al. (2022) and add a MaxOut unit to every B-cos transformation. This, of course, doubles the number of parameters, which can skew the comparison to the baseline models. In the following, we assess how MaxOut impacts the model performance.

In particular, in Fig. B5 (a), we compare the performance of B-cos ViTs as presented in the main paper to a version that does not use MaxOut when converting the baseline Transformer layers. We

---

[6]As was shown in Böhle et al. (2022), higher values of B can lead to a higher degree of alignment, but this transition is smooth and a value of B=1 in the attention layers seems to be sufficient.

observe that for smaller models (see, e.g., Tiny), MaxOut indeed significantly improves the performance of the B-cos ViTs. However, with increasing model size (Small and Base), this performance gap closes and the B-cos ViTs without MaxOut perform on par with those that have twice the number of parameters in the Transformer layers.

As a result, we find that for sufficiently large Transformers (Small and Base), the B-cos ViTs without MaxOut are able to achieve similar performance as the baseline models, without increasing the parameter count, as we show in Fig. B5 (b).

## C    IMPLEMENTATION DETAILS

In the following, we provide further implementation details regarding the models (C.1), the training and evaluation procedure (C.2), the explanation methods (C.3), and the evaluation metrics (C.4).

### C.1    MODELS

For all models, we rely on the implementation by Chefer et al. (2021), which we use unchanged for the conventional ViTs and modify as we describe below for the B-cos ViTs (C.1.1). The configurations of the ViTs follow the conventional specifications for ViTs of size Ti, S, and B, cf. Chefer et al. (2021). As described in Sec. 3, we use average pooling over the tokens and pass the result to the classifier head for all models. The conventional ViTs use a DenseNet-121 backbone as available in the torchvision library (Paszke et al., 2019).

### C.1.1    B-COS VITS

**Tokenisation.** For all B-cos ViTs, we use a pretrained[7] B-cos DenseNet-121 as provided by Böhle et al. (2022) as a tokenisation module, cf. Fig. 2a; specifically, we use the DenseNet-121 with the training[+] schedule, as this one achieves the same accuracy on the ImageNet validation set as the conventional DenseNet-121 contained in the pytorch library (Paszke et al., 2019). As described in the main paper, we freeze this backbone and extract features after either 13, 38, or 87 convolutional layers.

We then apply a single B-cos convolutional layer with a kernel size $k=1, 2, 4$ and stride $s=1, 2, 4$ with no padding on the feature maps after 87, 38, or 13 convolutional layers respectively, such that the number of tokens is the same for all B-cos ViTs. We found it advantageous to scale the features of the backbones by $10^3$, as this improved signal propagation and lead to better results[8]. Depending on the size of the transformer (Ti, S, B, see main paper), this B-cos convolution produced activations with $c=192, 384, 768$ channels after MaxOut (Goodfellow et al., 2013) over every two output units. The resulting activation map of size $c \times h \times w$ is then reshaped to $n \times c$ with $n$ the number of input tokens.

**Attention.** As described in the main paper, we replace the value computation as well as the linear projection of the attention heads by linear B-cos transformations. Further, we apply layer normalisation to the inputs before the query and key computations; the value computations use the raw input, cf. Fig. 2a. Finally, when additionally learning 'attention priors', see Sec. 3.5, we add a learnable parameter $\mathbf{B} \in \mathbb{R}^{m \times n \times n}$ to each attention layer, with $m$ the number of attention heads. The parameter $\mathbf{B}$ thus contains separate pair-wise priors between any two tokens for every attention head.

**MLPs.** As discussed in Sec. 3.4, we convert the MLPs to B-cos MLPs by replacing the linear transformations by B-cos transformations with two units and MaxOut (Goodfellow et al., 2013). Further, we remove the normalisation layer before the MLP block, and the GELU (Hendrycks & Gimpel, 2016) non-linearities within the MLPs.

**Classifier.** We use a single B-cos transformation as a classification head, without MaxOut and $C=1000$ output features, i.e., one output for each class.

**General remarks.** Similar to Böhle et al. (2022), we scale the output of every B-cos layer in the network by a scaling factor $\gamma = f/\sqrt{c}$ to improve signal propagation. In particular, as more channels lead to a stronger decay, we used $f=15, 20, 25$ for ViTs of size Ti, S, B respectively; further, since the multiplicative prior computes the product of two attention values $\leq 1$, the activations in these networks decay even more quickly and we scale each $f$ by an additional factor of 10. Moreover, as

---

[7]The pretrained models were downloaded from github.com/moboehle/B-cos.

[8]Note that in contrast to standard ViTs, in which normalisation layers ensure that the input to each layer is well-behaved, in B-cos Networks the activations can decay very quickly since no normalisation is used.

in Böhle et al. (2022), we scale down the model output by $10^3$ after which we add a logit bias $\mathbf{b} \in \mathbb{R}^C$ to the model output which is set to $\log(0.01/0.99)$ for each of the $C$ output logits. Lastly, for the B-cos ViTs, we encode the input images as in Böhle et al. (2021), i.e., such that each pixel uses 6 color channels $[r, g, b, 1{-}r, 1{-}g, 1{-}b]$ with $r, g, b \in [0, 1]$. As we discuss in Sec. C, this allows for visualising the matrices $\mathbf{W}(\mathbf{x})$ in color.

### C.2 Training and Evaluation Procedure

**Training.** We trained the B-cos ViTs with a batch size of 256. For the conventional ViTs, we found larger batch sizes to yield better results and thus trained those with a batch size of 1024. Further, we trained all our models with RandAugment (Cubuk et al., 2020) ($n{=}2$ and $m{=}9$) and used images of size $224{\times}224$. While the conventional models were, as is common, trained with SoftMax and a cross entropy loss, for the B-cos ViTs we followed Böhle et al. (2022) and trained with binary cross entropy and sigmoid applied to the output logits. Note that, as discussed in Böhle et al. (2022), binary cross entropy induces the necessary logit maximisation for every input, which in turn leads to weight alignment with class-relevant patterns.

**Evaluation.** We evaluated all networks on the ImageNet validation set after resizing the images such that the smaller dimension measured 256 pixels and then center-cropped images of size $224{\times}224$.

### C.3 Attribution Methods

In the following, we describe the explanation methods that we evaluate on the B-cos as well as the conventional ViTs in more detail.

**Model-inherent explanations.** The model-inherent explanations that we quantitatively evaluated are given by the contribution maps defined in Eq. (7). Specifically, for a given class logit, we extract the effective linear contribution as performed by the model and multiply it with the input in an element-wise manner and sum all values per pixel location, i.e., across the colour channels; note that this is conceptually equivalent to 'Input$\times$Grad' for piece-wise linear models. For a visualisation of contribution maps, see Figs. 1 and 7; here, we use a blue-white-red colormap with blue colors denoting negative, and red colors denoting positive contributions. This visualisation method is the same for all methods except for those that use the jet colormap (CheferLRP, Rollout, and FinAtt). Additionally, for better visibility, we clamp the contribution values to the interval $[-v, v]$, with $v$ the 99.9th percentile of the absolute values of the given spatial attribution map (i.e., after summing over the colour channels). Note that as a result, the explanations are normalised *independently*; for a discussion of the effect of normalising across multiple explanations, see Appendix A.2.

Further, as also shown in those figures, the B-cos formulation allows to directly visualise the transformation matrix $\mathbf{W}(\mathbf{x})$ in color. Specifically, note that the input to B-cos networks is encoded as $\mathbf{p} = [r, g, b, 1{-}r, 1{-}g, 1{-}b]$ with $r, g, b \in [0, 1]$ the color channels, cf. Sec. C.1.1 and Böhle et al. (2022). As such, the color of each pixel is *unambiguously* encoded by the angle of the pixel vector $\mathbf{p}$ and it is thus possible to reconstruct the image colors from the angles alone. Crucially, as discussed in Sec. 3.1 $\mathbf{W}(\mathbf{x})$ is implicitly optimised to align with relevant patterns in the input, i.e., to have a similar angle as the input, such that the weights for any given pixel can be mapped to a specific color in RGB. We further follow Böhle et al. (2022) and use the *norm* of the pixel vectors to compute the opacity $\alpha$ in an RGB-$\alpha$ encoding, for details see Böhle et al. (2022), and only show pixels that *positively* contribute to the respective logit.

**Transformer-specific explanations.** As described in the main paper, we evaluate against common transformer-specific explanations such as the average attention distribution in the final layer (FinAtt), Attention Rollout as proposed by Abnar & Zuidema (2020), and GradSAM (Barkan et al., 2021). On the conventional ViTs, we further evaluate LRP-based explanations: partial LRP (pLRP) Voita et al. (2019) and the transformer-specific LRP adaptation by Chefer et al. (2021) (CheferLRP). For all these transformer-specific explanations, we rely on the implementation provided by Chefer et al. (2021).

**Architecture-agnostic explanations.** Further, as shown in Figs. 6 and 7, we also evaluate other commonly used explanation methods. Specifically, we use IntGrad (Sundararajan et al., 2017) with $n{=}32$ steps, 'Input$\times$Gradient' (IxG), cf. Adebayo et al. (2018), as well as an adapted Grad-CAM (Selvaraju et al., 2017) as in Chefer et al. (2021). For the last, we rely on the implementation provided by Chefer et al. (2021). For the remaining methods, we use the implementations contained in the *captum* library of pytorch (Paszke et al., 2019).

## C.4   EVALUATION METRICS

### C.4.1   LOCALISATION METRIC

For the localisation metric, we evaluated all attribution methods on the *grid pointing game* (Böhle et al., 2021). For this, we constructed 250 $2 \times 2$ grid images, see, e.g., Fig. 4. As was done in Böhle et al. (2021), we sorted the images according to the models' classification confidence for each class and then sampled a random set of classes for each multi-image. For each of the sampled classes, we then included the most confidently classified image in the grid that had not already been used in a previous grid image.

Further, transformers with positional information expect a fixed size input, see Sec. 4. To nevertheless evaluate the attribution methods on the localisation metric, which uses images of size $448 \times 448$, we scale the synthetic images down by a factor of two such that they are of size $224 \times 224$ and thus of a size that is compatible with the transformers.

### C.4.2   PERTURBATION METRIC

As for the pixel perturbation metrics, we proceed as follows.

First, we sort the images in the validation set by their classification confidence and evaluate on the first 250 images for each model. We do this to reduce the computational cost of evaluating this metric whilst nevertheless allowing for a fair comparison between models. Specifically, we observed the model stability to correlate with the model confidence and by choosing the most confidently predict images, each model is evaluated in a favourable setting, which ensures comparability.

Second, we remove up to 25% of the pixels from the images in increasing / decreasing order as ranked by a given attribution method. Specifically, we sample the resulting confidence curves at 9 equidistant points $r$ in the interval [0, 25%] and 'remove' the pixels by zeroing out the respective pixel encoding.

Finally, we record the mean model confidence at the sampled points, which we normalise by the initial mean confidence; as such, each curve starts at $(r, o) = (0, 1)$ with $r$ the percentage of pixels removed and $o$ the normalised model confidence. To assess the quality of the ranking, we then measure the area between the two confidence curves corresponding to the order in which we remove the pixels, i.e., least / most important first, see, e.g., Fig. 6 (right).

