# OpenReview forum: "Holistically Explainable Vision Transformers"
_ICLR.cc/2023/Conference — Submitted to ICLR 2023_

### Official Review · Reviewer_rHTt · 2022-10-21

**Confidence:** 4
**Correctness:** 3
**Technical Novelty And Significance:** 2
**Empirical Novelty And Significance:** 2
**Recommendation:** 5

**Clarity, Quality, Novelty And Reproducibility:**

The overall quality of this paper is good and the technical part is novel. The authors have not provided the code base in terms of reproducibility.

**Strength And Weaknesses:**

Strength:

•	The qualitative results look promising. The proposed method generates visualizations with richer details than the baseline methods shown in Fig. 7 and Fig. A1.
•	The quantitative results look impressive. The proposed B-cos Transformer have higher localisation and perturbation scores than the baseline methods under the same model accuracy. Compared to the conventional ViT models of the same depth, B-cos Transformer has better overall performance across different depths.
•	The paper is well-written and easy to follow.

Weakness:

•	The authors argue that B-cos Transformer performs at least as well as the baseline ViTs under the same number of backbone layers in Section 5.1. However, the experimental setting is unfair without comparing the computational complexity to the baseline methods, especially when the authors replace the conventional tokenization module with a pre-trained B-cos DenseNet-121. Though deriving more explainable models, the proposed model should be toned down if they are too heavy to deploy. This is the same for Fig. 6, which also lacks comparisons of model complexity vs. Localisation/perturbation.

•	I am also confused about where the good performance of B-cos Transformer comes from. The authors claim that the positional information design largely improves performance. However, the effectiveness of the positional information cannot be concluded from Fig. 5, with many other components modified. To fully explain the performance gain, comparisons have to be done by attaching the B-cos positional information to the conventional ViTs.

•	I am also curious about how the third column of Fig. 7 is generated, as the visualizations do not seem to be normalized.


**Summary Of The Paper:**

The paper proposes a new Transformer architecture for better holistic explainability. The architecture generates explanations that reflect all the model components instead of only the attention layers. To do so, the authors take explicit designs for each of the Transformer modules so that they can be summarized by a single linear transformation. The authors perform the main and ablative experiments on ImageNet to evaluate the effectiveness of the proposed methods.

**Summary Of The Review:**

There are many confusing points that need to be addressed.

---

> ### Author Response · Authors · 2022-11-11
> **General response reviewer rHTt**
>
> We thank the reviewer for the time taken to provide detailed and constructive feedback to our work. We are very encouraged to find the reviewer to deem our qualitative results to be "promising" and the quantitative results to be "impressive", and appreciate that the reviewer positively highlights the presentation of our work.
>
> In the following three comments, we would like to address the reviewer's remaining concerns. In particular, we address the concern regarding the model complexity (Weakness 1) as well as the role of the multiplicative attention with respect to the model performance (Weakness 2). Finally, we describe in detail how the model-inherent explanations are generated (Weakness 3); we updated the manuscript to state this more clearly and thank the reviewer for pointing out that this was unclear.

---

> ### Author Response · Authors · 2022-11-11
> **Weakness 1 — Model complexity**
>
> We believe this concern to be based on two misunderstandings and clarify in the following.
>
> First, please note that all models (i.e., B-cos ViTs *and* conventional ViTs) feature a pre-trained convolutional backbone. Concretely, as we describe in the "Models" paragraph in section 4, in the case of the B-cos ViTs we use a **pretrained B-cos DenseNet-121** and for the conventional ViTs we use a **standard pre-trained DenseNet-121** (as obtained from the torchvision library); we chose these models since they achieve the same accuracy on the ImageNet test set (see Boehle et al., 2022), which thus allows for a fair comparison. As such, all models use the same architectural design and the (B-cos / conventional) Transformers are applied to the pre-trained features of the respective pre-trained backbone extracted after 13, 38, or 87 layers as indicated by the colour coding in Fig. 5.
>
> Secondly, note that Fig. 6 only contains results of B-cos ViTs and is designed to compare the quality of the shown attribution methods (Ours, Rollout, GradSAM, etc.) when evaluated on the exact same model (the X-axis shows the respective accuracy of each model); additional comparisons between explanations for B-cos ViTs and conventional ViTs are provided in the supplement.
>
> To put it simply, the main message of Fig. 6 is that the model-inherent explanations (blue points) consistently obtain significantly higher localisation scores than all tested post-hoc explanations methods (other colours) for every single model (X-axis). This shows that the model-inherent contribution maps lend themselves better for explaining the models than all of the tested post-hoc explanation methods.
> Since the model complexity is thus the same for every explanation method (they are evaluated for the same models) and the goal of this figure is to compare explanation methods for the *same models*, we are unsure what the purpose of explicitly including the model complexity in Figure 6 would be and would highly appreciate it if the reviewer could clarify.
>
> Finally, while the general architecture of all models is thus comparable (pre-trained DenseNet backbones on which Transformers are trained), there is, of course, nonetheless a difference in computational complexity between the models B-cos ViTs and the conventional ViTs. For details, please also see the answer to reviewer oHsW.

---

> ### Author Response · Authors · 2022-11-11
> **Weakness 2 — The role of multiplicative attention**
>
> We fully agree with reviewer 3 that Fig. 5 cannot serve as a basis for attributing any observed performance gains to the multiplicative attention prior alone, it was not our intention to suggest otherwise; if there is a particular phrasing that caused this confusion, we would be very grateful if the reviewer could point this out and we will update the manuscript accordingly.
>
> The conclusions that we *did* draw from our results are the following:
>
> **(1) B-cos ViTs benefit from multiplicative attention priors.**
> This conclusion is based on the experimental results shown in Fig. 3 in which we plot the test accuracy of B-cos ViTs of various sizes trained with different strategies w.r.t. how to provide the positional information. As can be seen, the multiplicative attention leads to significant and consistent gains; i.e., B-cos ViTs with multiplicative attention perform better than B-cos ViTs with the conventional positional embedding or the additive attention described in Eq. (18).
>
> **(2) B-cos ViTs only perform on par with conventional ViTs if trained with multiplicative attention priors.**
> Note that the results in Fig. 3 for the differently sized ViTs can be compared with the respective dark red bars in Fig. 5, i.e., for the ViTs with a backbone of size 87. As becomes clear, the standard positional embedding (blue bars, Fig. 3) and the additive attention prior (yellow bars, Fig. 3) consistently perform worse than conventional ViTs of similar sizes (dark red bars of "Conventional" in Fig. 5 for Tiny, Small, and Base). Only with the multiplicative attention priors (green bars, Fig. 3) do the B-cos ViTs perform on par with the conventional ViTs (see respective dark red bars for "Conventional" and "B-cos" in Fig. 5). For clarity, we summarise those results in the following table:
>
> |$\downarrow$ Model size $\vert$ Model Type $\rightarrow$  |B-cos (Pos. Embed) | B-cos Add. Att.  | B-cos Mul. Att.  | Conventional ViT  |
> |:---|:---:|:---:|:---:|:---:|
> |  Tiny |  71.0 | 71.5  | **75.6**  |  74.6 |
> |  Small | 74.3  | 74.3  |  **76.6** | 74.9  |
> |  Base  |  73.1 | 73.9  | **76.7**  | 74.5  |
>
>
> **(3) Conventional ViTs do not seem to benefit from multiplicative attention.**
> As described in 5.1, we did not find similarly consistent performance gains when training conventional ViTs with the multiplicative attention prior, as suggested by the reviewer, see the following results:
>
> |$\downarrow$ Model size $\vert$ Attention Type $\rightarrow$  | Pos. Embed | Add. Att.  | Mul. Att. |
> |:---|:---:|:---:|:---:|
> |  Tiny |  **74.91** | 74.28  | 74.63  |
> |  Small | 74.77  | **74.82** |   74.76|
> |  Base  |  **74.40** | 74.28  | 74.26 |
>
> As can be seen, the conventional ViTs perform similarly across the different strategies for providing positional information. We will add these results to the final version for completeness.

---

> ### Author Response · Authors · 2022-11-11
> **Weakness 3 — Generation of the attribution maps**
>
> To visualise the contribution maps (Ours Eq. (7)) in Fig. 7, as well as those of GradSAM, GradCAM, IxG and IntGrad, we proceeded as follows; we updated section C.3 in the appendix to reflect this more clearly.
>
> **First**, we compute the respective importance attribution to input colour channels (see, e.g., Eq. 7 for 'Ours') to the respective class logit ('black swan', 'forklift', etc.).
>
> **Second**, as is common practice, we sum the values across the colour channels to obtain a single value per pixel location.
>
> **Third**, each map is individually plotted with a blue-white-red colour map over the interval $[-v, +v]$, with $v$ the 99.9th percentile of the absolute values in the given contribution map.
>
> As such, the reviewer is right in observing that the visualisations in the third column of Fig. 7 are currently not normalised across images; however, as is common practice, neither are those of the other attribution maps.
>
> It is, however, of course possible to normalise the contribution maps across images. In fact, as the sum over the contribution maps is equivalent to the class logit produced by the model (up to a single, fixed bias term b, see "General remarks" in section C.1.1 in the appendix) for the model-inherent contribution maps, such a normalisation would additionally faithfully reflect the model decision in terms of relative logit scores between different explanations on the same image. We chose not to normalise the contribution maps in favour of better visibility of the explanations, especially for IntGrad. For the model-inherent contribution maps, the observed differences are minor as long as the class logits are on a similar scale, see section A.2 in the updated supplement (changes are marked in colour).

---

### Official Review · Reviewer_NoVr · 2022-10-24

**Confidence:** 4
**Correctness:** 2
**Technical Novelty And Significance:** 2
**Empirical Novelty And Significance:** 3
**Recommendation:** 3

**Clarity, Quality, Novelty And Reproducibility:**

The idea of extending the Bcos Networks to transformer is new, but could be incremental. The presentation and writing are good.  The link of the code cannot be opened.

**Strength And Weaknesses:**

Strength:
+ The idea of improving interpretation of transformer is interesting.
+ The paper is well organized and easy to follow.
+ The method is well evaluated and the results are well presented.

Weakness:
- My first concern is the claim of Bcos-ViTs. The authors are using transformers on the top of DenseNet features which is not ViT. A ViT is trained directly on images by considering each patch as a token. The method is more like Bcos Transformer, not Bcos-ViT.
- The whole explanation principle is exactly the same as the Bcos Networks (CVPR2022) which makes this paper more like an extension from deep network to transformer. What is the new challenge here for explaining DenseNet+transformer using Bcos idea as compared to explaining a DenseNet?
- The method itself cannot be adopted to explain existing ViT models like [Chefer et al. (2021)], but has to be re-trained to gain interpretation ability. But this paper does not show how the ViT would performance if all the transformers are replaced with the proposed block. It is very misleading to claim DenseNet+Transformer as ViT in Fig 5. How would a standard ViT perform if the proposed Bcos method is used?

**Summary Of The Paper:**

This paper proposes to replace the modules of transformer with dynamic linear operation so that it is more explainable. It follows the guideline of BCos Networks but adopts transformers on the top of DenseNet features. The method is evaluated on ImageNet with localization and perturbation metric. It claims to have to be more interpretable and perform competitively to ViT.

**Summary Of The Review:**

Overall, I feel that the contribution does not meet the bar of ICLR and some claims are very misleading to readers. I would recommend weak reject for the current version.

---

> ### Author Response · Authors · 2022-11-11
> **General response to reviewer NoVr**
>
> We thank the reviewer for the time taken to provide detailed and constructive feedback to our work. We are encouraged to find the reviewer to positively highlight our experimental evaluation and the organisation of our manuscript.
>
> In the following three comments, we would like to address the reviewer's remaining concerns. In particular, we address the concern regarding the naming of our method (Weakness 1) and our technical contributions (Weakness 2). Finally, we discuss how the model-inherent explanations compare to post-hoc explanations and respond to the reviewer's concern that the presentation of our results could be misleading (Weakness 3).

---

> ### Author Response · Authors · 2022-11-11
> **Weakness 1 — Method name**
>
> We apologise for the perceived ambiguity in the naming of our proposed models and fully agree with the reviewer that the usage of the term "ViT" in the literature is inconsistent—it is both used to describe specific architectures, as well as an entire family of models. We are happy to include a detailed discussion in our work and to clearly define how we use the term ViT; if deemed appropriate and the reviewers as well as the area chair agree, we are also open to changing the name.
>
> That said, misleading the readers was certainly not our intention in naming the proposed models “B-cos ViTs”. Instead, with this we followed the convention in the literature to include “ViT” in architecture names derived from the original ViTs. This is also the case, if the respective models use a convolutional backbone for tokenisation, see, e.g.:
>
> - Xiao et al. (Neurips, 2021) increase the depth of the convolutional stem and nonetheless call the models ViTs.
> - Yuan et al. (ICCV, 2021) introduce the T2T ViT, which features a convolutional backbone and includes “ViT” in the model name.
> - Graham et al. (ICCV, 2021) introduce LeViT, which features a convolutional backbone and includes “ViT” in the model name
> - Zhang et al. (ICCV, 2021) introduce ViT-YOLO, which features a convolutional backbone and includes “ViT” in the model name.
>
> The naming of our models simply follows this practice.
>
> In fact, we believe “B-cos ViT” to more adequately describe our experimental setting than “B-cos Transformer” as it is the less general name. Specifically, please note that the ViTs (Dosovitskiy et al., 2021) use the exact same architecture as the original Transformer (Vaswani et al, 2017) with model configurations as used in the BERT models (Devlin et al., 2019). As such, ViTs and Transformers mainly differ in the type of input and the modality-specific tokenisation module. Since we apply our B-cos Transformers to *vision* input, we believe B-cos *Vision* Transformers (B-cos ViTs) to more clearly signal to the reader on which domain our models are evaluated and which type of Transformer we analyse.
>
> In summary, we would like to politely rebut the assertion that our claim is inaccurate or misleading. Nonetheless, we are of course open to renaming our proposed method as deemed appropriate.

---

> ### Author Response · Authors · 2022-11-11
> **Weakness 2 — Technical contributions**
>
> We would like to highlight that our adaptation of the B-cos framework follows from a detailed analysis of the individual components, and, where necessary, implements substantial changes to the ViT architecture.
>
> For example, the B-cos models by Boehle et al. (2022) do not contain any **(1)** normalisation layers, **(2)** attention layers, or **(3)** positional encodings, which each need to be addressed adequately to maintain performance and interpretability.
>
> **For (1)**, we move the layer normalisation inside of the attention computation, as described in section 3.3. This allows for taking advantage of the layer norm benefits (standardising the input to the softmax function in the attention layer), whilst ensuring that the ViT model still computes its output according to the requirements of the B-cos framework. Additionally, as we show in section B.2 in the supplement, this significantly improves the model performance.
>
> **For (2)**, to the best of our knowledge our work is the first to analyse and discuss in detail how to combine attention maps with the B-cos-based linear mappings. In detail, we show that treating the attention maps themselves as a dynamic linear mapping allows for integrating them seamlessly into the B-cos-based explanations. While this choice might seem intuitive in hindsight, we would like to point out that this has not been done before.
>
> **For (3)**, we introduce a novel form of providing positional information to the ViT models, which significantly improves the performance of the B-cos ViTs. Importantly, this approach not only improves performance, but also benefits the model interpretability, since the conventional positional embedding adds a bias term to the input that would not be accounted for in the visualisations of the model-inherent explanations. With the multiplicative embedding, we ensure that the model output is exactly given by the sum over the contribution maps as shown in Fig. 7 (col. 3).
>
> In short, we would like to point out that our work on B-cos ViTs is not a trivial extension. The compatibility of the ViTs with the B-cos framework and the strong performance of the B-cos ViTs  are not a priori to be expected and our work thus constitutes an important data point for other researchers to refer to, as it includes a thorough analysis of and a detailed comparison to conventional ViT-based models under various settings.

---

> ### Author Response · Authors · 2022-11-11
> **Weakness 3 — The method cannot explain conventional ViTs**
>
> **B-cos interpretability requires re-training**
>
> The reviewer is completely right in highlighting that designing inherently interpretable models requires training new models—this, however, is true for any novel DNN architecture and an unavoidable part of research and design of inherently interpretable models.
>
> Of course, model-inherent explanations thus stand in contrast to post-hoc explanations for pre-trained models such as, e.g., Chefer et al. (CVPR, 2021), which also constitute an important research direction on increasing DNN interpretability.
>
> While it would be ideal to obtain faithful and interpretable post-hoc explanations of today's models, we would like to note that it is still an open question how to achieve this in a reliable manner. In this paper, we therefore follow a different route: in particular, and similar to Rudin (Nature Machine Intelligence, 2019) we believe that an alternative—and potentially in the long run more powerful—route is to include the goal of interpretability already during the design and optimisation process of the next generation of models. On this route, our work makes an important step towards establishing highly performant interpretable models in the community.
>
> Lastly, please note that these models would also need to be trained only once, similar to conventional models, but additionally provide highly detailed and, importantly, model-faithful explanations for their decisions that are unprecedented in this form for Vision Transformers. Since we will open-source all our models and our code upon publication, these pre-trained interpretable models will be available for everyone to use.
>
>
> **Misleading claim**
>
> As discussed in the answer to the reviewer’s first point of concern (Weakness 1), it was certainly not our intention to mislead the readers with our naming convention and we are open to changing the name as deemed appropriate.
>
> That said, we are convinced that we fully and transparently described our experimental protocol in detail and in a clear manner. Figure 5 very clearly indicates the number of backbone layers for every single model and we therefore politely object to the assertion that the figure is misleading.
>
> We are further convinced that our results clearly support our claim: in particular, for all the tested conventional ViT architectures (with the described backbones) it is possible to maintain performance (see Fig. 5) whilst significantly increasing their interpretability (see Fig. 6) by converting them to B-cos ViTs as described in section 3. With this, we provide an important contribution towards establishing performant and interpretable models as a viable alternative to conventional non-interpretable models.
>
>
> **Additional comments**
>
> Lastly, we would like to ask the reviewer whether the score for the paper is as intended (in the review summary, the reviewer recommends a "weak reject", but the official score given is a "reject") and apologise for the confusion regarding the link to the code — "Code will be available: github.com/anonymous/authors" was meant to be a placeholder during the review period highlighting our commitment to making our results fully reproducible and open-sourcing our code.

---

### Official Review · Reviewer_oHsW · 2022-10-25

**Confidence:** 4
**Correctness:** 4
**Technical Novelty And Significance:** 3
**Empirical Novelty And Significance:** Not applicable
**Recommendation:** 6

**Clarity, Quality, Novelty And Reproducibility:**

The paper was very well written and was very easy to follow. The authors mentioned that the code will be released upon publication. The architecture is an extension of the model proposed in "B-cos Networks: Alignment is All We Need for Interpretability" and seemed a bit derivative.

**Strength And Weaknesses:**

Pros
	- The proposed method is simple and seems easy to implement with the provided details.
	- Strong benchmark performance improvements across multiple explainability metrics. Further, the attribution maps of the model are highly detailed and have very low noise.

Cons
	- This paper is a direct extension of "B-cos Networks: Alignment is All We Need for Interpretability" by Bohle et al, which proposed the B-cos transform and implemented it on MLPs and CNNs. In this paper, that transform is applied to the vision transformer. However, the main contribution of this paper is incorporating the B-cos transform in the attention layers and the positional encodings. It's incorporation in the MLP/CNN layers is directly taken from the original paper. This makes the architecture seem a bit derivative.
	- The paper proposes using max-out network in equation 12. However, that would double the number of parameters in the model. It would be helpful if some comments are added regarding the increased model size. It would also be helpful to know if removing the max-out layer will significantly impact the final results.
Most of the results compared the model with the state-of-the-art visualization models. Given this model is a significant architectural modification of the original vision transformer, it would be really helpful to compare the results of the model with those of the vision transformer on other benchmark tasks.

**Summary Of The Paper:**

The paper proposes a novel approach towards improving the explainability of vision transformers. Specifically, the paper proposes updating each component in the model using the B-cos transform, which is designed to increase the alignment of the inputs and the weights. The paper builds on a previous state-of-the-art work that introduced the B-cos transform on DNN/CNNs and extends it to vision transformers. Comprehensive experiments are conducted comparing the proposed approach with several state-of-the-art explainability methods.

**Summary Of The Review:**

I don't think it's a strong accept due to the slightly derivative nature of the work, but overall, the paper extends a very novel idea from DNN/CNNs to vision transformers and supports this with strong benchmark results. Due to this, I think its marginally above the acceptance threshold.

---

> ### Author Response · Authors · 2022-11-11
> **General response to reviewer oHsW**
>
> We thank the reviewer for the time taken to provide detailed and constructive feedback to our work. We are encouraged to find our quantitative ("Strong benchmark performance") and qualitative results ("highly detailed and very low noise") to be highlighted so positively, and appreciate that our paper is deemed "very well written".
>
> In the following two comments, we would like to address the reviewer's remaining concerns regarding the novelty of our work (Weakness 1) as well as the parameter count of the B-cos models and the extension to other tasks (Weakness 2).

---

> ### Author Response · Authors · 2022-11-11
> **Weakness 1 — Novelty**
>
> While the proposed solution might be perceived as obvious in hindsight, we would like to point out that it is not a priori clear how to combine ViTs and the B-cos framework. Doing so requires a detailed analysis, which we perform in this work. Based on this analysis we show how to integrate the idea of the B-cos transformations into Vision Transformer architectures and implement substantial changes where necessary. In particular, we show how to properly merge the attention maps with the B-cos-based explanations, discuss how positional information can be made available to B-cos ViTs to maintain performance and interpretability, as well as change the position of the normalisation layer; for an ablation regarding the latter, see section B.2 in the supplement.
>
> For an additional, more detailed discussion regarding the technical contributions of our work, **please also see the answer to reviewer NoVR (Technical contributions).**
>
> As a result, and to the best of our knowledge, our work thus introduces the first inherently interpretable ViT architecture and shows that the high performance of these models can be maintained whilst additionally endowing them with a high degree of inherent interpretability.
>
> Additionally, given the holisticness of the model-inherent explanations, we are able to clearly highlight shortcomings of the widely used attention-based explanations, which we believe to be another important contribution. In particular, when comparing the model-inherent explanations of the B-cos ViTs and, e.g., attention rollout (see equation (A.2) in the supplement), it becomes clear how attention rollout neglects many potentially important network components in the model explanations and cannot give class-specific explanations (see, e.g., Fig. 7).

---

> ### Author Response · Authors · 2022-11-11
> **Weakness 2 — Parameter count and other tasks**
>
> **Regarding the impact of MaxOut**
>
> The reviewer is right in remarking that B-cos models have a higher parameter count than conventional models due to the usage of MaxOut, which doubles the number of the respective weight parameters. With this design choice, we followed the recommendations of Boehle et al. (CVPR, 2022) with respect to the conversion of conventional model layers to B-cos layers. We fully agree with the reviewer, however, that this point deserves additional discussion. Judging from the results without MaxOut reported in Boehle et al. (CVPR, 2022), we expect a drop of maximally 1-2p.p. in top-1 accuracy, which is in line with preliminary results we observe for the requested experiments. In fact, we see that while for smaller models (B-cos ViT-S) there is indeed a slight drop in accuracy (B-cos ViT-S-13), the performance gap closes for larger ones (B-cos ViT-S-87 and B-cos ViT-B-13) as we report in the following table:
>
> |$\downarrow$ Model type $\vert$ Model config $\rightarrow$  |B-cos ViT-S-13 | B-cos ViT-S-87  | B-cos ViT-B-13  |
> |:---|:---:|:---:|:---:|
> |  B-cos $\vert$ w/ MaxOut |  **74.2** | **76.6**  | 74.9|
> |  B-cos $\vert$ no MaxOut | 72.7  | 76.1  |  **75.3**|
> |  Conventional ViT  |  73.0 | 74.9  | 73.1 |
>
> We will include the full set of results for models without MaxOut to the final version if deemed appropriate and will extend the discussion accordingly. Please note that the Transformers of the B-cos ViTs without MaxOut even have fewer parameters than their conventional counterparts (fewer bias parameters), but still achieve comparable classification accuracies.
>
> **Evaluating B-cos ViTs on other tasks**
>
>
> We are convinced that introducing the first inherently interpretable Vision Transformer is in and of itself an important milestone towards establishing interpretable models as a viable alternative to conventional black-box models and will be crucial for bringing interpretable deep learning more into the focus of our community and thus spur additional research regarding XAI.
>
> Nonetheless, we fully agree with reviewer oHsW that there are still open questions that have to be addressed in future work, e.g., evaluating the B-cos ViTs on other tasks and datasets. In this work, however, we decided to instead focus on a comprehensive and in-depth evaluation in the classification setting and provide a detailed discussion of our proposed method. For this, we introduced and evaluated different attention mechanisms on various model configurations (Fig. 3),  compared a wide range of baseline models to their respective B-cos counterparts (Fig. 5), and analysed the interpretability of the respective models in detail.
>
> Since we will open-source all our models and our code, our interpretable B-cos ViTs will be available for everyone to use and extend to other tasks and datasets.

---

> > ### Author Response · Authors · 2022-11-17
> > **Weakness 2 — Update**
> >
> > In the following table, we report the full results after training all ViT models without MaxOut. As discussed above, MaxOut seems to be particularly beneficial for smaller models (e.g., ViT-Ti-13 performance: 66.9 $\rightarrow$ 71.1). However, for larger models, we find that the performance gap closes (see ViT-B-{13, 38, 87}) and the models without MaxOut perform on par with those with MaxOut. We updated the manuscript to reflect these results, please see section B.3 and Figure B5 in the updated appendix.
> >
> >
> > | $\downarrow$ Model config $\Vert$ Model type $\rightarrow$ | B-cos no MaxOut | B-cos w/ MaxOut | Conventional ViT |
> > | :---------------------------------------------------------- | :-------------------------: | :-----------------------: | :----------------: |
> > |  ViT-Ti-13  |  66.9  |  71.1  |  **71.4**  |
> > |  ViT-Ti-38  |  70.4  |  **74.3**  |  72.8  |
> > |  ViT-Ti-87  |  73.7  |  **75.6**  |  74.6  |
> > |  ViT-S-13  |  72.7  |  **74.2**  |  73.0  |
> > |  ViT-S-38  |  74.6  |  **75.7**  |  74.5  |
> > |  ViT-S-87  |  76.1  |  **76.6**  |  74.9  |
> > |  ViT-B-13  |  **75.3**  |  74.9  |  73.1  |
> > |  ViT-B-38  |  76.4  |  **76.6**  |  74.3  |
> > |  ViT-B-87  |  **76.8**  |  76.7  |  74.5  |

---

### Author Response · Authors · 2022-11-17
**Common Response to All Reviewers**

We would again like to thank the reviewers for their time and the valuable feedback on our submission. We updated our manuscript based on the given suggestions and marked the changes in colour (sections 5.1, A.2, B.3 and C.3).

If we adequately addressed the reviewers' concerns and there are no further questions, we would kindly ask the reviewers to consider raising their score to reflect this. If there are indeed still open questions, we would be happy to answer any additional questions that remain.

Sincerely,
Paper 1636 Authors

---

### Decision · Program_Chairs · 2023-01-20

**Decision:**

Reject

**Justification For Why Not Higher Score:**

- Incremental work with respect to (Böhle et al., 2022).
- No analysis of how the explanation can shade light on the difference between ViT vs ConvNets predictions
- Consolidated experiments

**Justification For Why Not Lower Score:**

N/A

**Metareview: Summary, Strengths And Weaknesses:**

This paper introduces the B-cos Vision image Transformer (ViT), which is explainable by design. The method consists in adapting the B-cos models for ConvNet to transformers networks, especially by adapting the approach to take into account normalisation layers, attention layers, and positional encodings. Experiments have been conducted on ImageNet, and show that B-cos ViTs perform on par with ViTs, while providing a more accurate explanation.
The paper initially received one reject (3), one borderline reject (5), and one borderline accept recommendation (6). The reviewers overall found the paper clear and well written, but their concern related to the novelty of the approach, clarifications on experiments, and on the mismatch in architecture between the B-cos ViT and vanilla ViT. The rebuttal brings additional results related to the impact of maxout, multiplicative attention, and statements on paper claims and contributions. After the rebuttal, all reviewers stuck on their initial grades.

The AC carefully reads the submission and discussions. The AC considers that the paper is clear and well presented, and that introducing an new explainable by design ViT is interesting even if it does not strictly follow the vanilla ViT architecture. However, the AC also considers that the B-cos ViT is a relatively direct adaptation of the B-cos ConvNets (Böhle et al., 2022). In addition, the provided explanation from B-cos ViT are qualitatively similar to their convolutional counterpart, and no comparison is provided to analyse if the explanation can reveal systematic differences between ConvNet and transformers models. Regarding B-cos ViT architecture, the specific impact of the multiplicative attention is very large while it is small for vanilla ViT, which calls for more investigations.
Therefore, the AC recommends paper rejection but highly encourages the authors to resubmit their work based on the reviews' feedback.